# Structures of lipoprotein signal peptidase II from _Staphylococcus aureus_ complexed with antibiotics globomycin and myxovirescin

Samir Olatunji [1,6], Xiaoxiao Yu[1,6], Jonathan Bailey[1,6], Chia-Ying Huang [2], Marta Zapotoczna [3], Katherine Bowen[4], Maja Remškar[5], Rolf Müller [5], Eoin M. Scanlan[4], Joan A. Geoghegan[3], Vincent Olieric [2] & Martin Caffrey [1]*

Antimicrobial resistance is a major global threat that calls for new antibiotics. Globomycin and myxovirescin are two natural antibiotics that target the lipoprotein-processing enzyme, LspA, thereby compromising the integrity of the bacterial cell envelope. As part of a project aimed at understanding their mechanism of action and for drug development, we provide high-resolution crystal structures of the enzyme from the human pathogen methicillin-resistant _Staphylococcus aureus_ (MRSA) complexed with globomycin and with myxovirescin. Our results reveal an instance of convergent evolution. The two antibiotics possess different molecular structures. Yet, they appear to inhibit identically as non-cleavable tetrahedral intermediate analogs. Remarkably, the two antibiotics superpose along nineteen contiguous atoms that interact similarly with LspA. This 19-atom motif recapitulates a part of the substrate lipoprotein in its proposed binding mode. Incorporating this motif into a scaffold with suitable pharmacokinetic properties should enable the development of effective antibiotics with built-in resistance hardiness.

[1] Membrane Structural and Functional Biology Group, School of Medicine and School of Biochemistry and Immunology, Trinity College Dublin, Dublin D02 R590, Ireland. [2] Swiss Light Source, Paul Scherrer Institute, CH-5232 Villigen, Switzerland. [3] Moyne Institute of Preventive Medicine, Department of Microbiology, School of Genetics and Microbiology, Trinity College Dublin, Dublin D02, Ireland. [4] School of Chemistry, Trinity College Dublin, Dublin D02 R590, Ireland. [5] Department Microbial Natural Products, Helmholtz-Institute for Pharmaceutical Research Saarland, Helmholtz Centre for Infection Research and Department of Pharmacy, Saarland University Campus E8 1, D-66123 Saarbrücken, Germany. [6]These authors contributed equally: Samir Olatunji, Xiaoxiao Yu, Jonathan Bailey. *email: martin.caffrey@tcd.ie

As we move into an era of precision medicine and rational use of therapeutics, the prospect of having available both broad-spectrum and species-specific anti-infectives is highly attractive. Early identification of the infectious agent and treatment with a therapeutic that killed or attenuated the pathogenic organism and no others would be the objective in the case of species-specific drugs[1]. In so doing, some of the detrimental consequences of using broad-spectrum antibiotics that include antibiotic-associated diarrhoea and thrush, where swaths of the microbiota are thrown out of kilter, would ideally be avoided[2]. Further, antibiotic resistance would be less of an issue since just the one species would endeavour to respond to the selective pressure. Anti-infective agents that reduce bacterial virulence without exerting direct bactericidal activity (as antibiotics do) have the added advantage of reduced selective pressure minimizing the risk of the emergence of resistant mutants.

One approach to species-specific drug development is through structure-based drug design. This requires the structure of at least one of the target orthologs, ideally at high resolution. However, having the structure of orthologs from other pathogenic species would prove extremely valuable. This is the goal of the current study, focussed on the lipoprotein signal peptidase II, LspA (abbreviations are included in Supplementary Table 1), an enzyme involved in lipoprotein posttranslational processing in bacteria (Fig. 1).

LspA is essential in Gram-negative bacteria, it has no mammalian equivalents and its active site is accessible to potential drugs at the outer surface of the inner membrane[3–5]. These features make it attractive as a drug target. Indeed, the natural antibiotics, globomycin and myxovirescin (Fig. 1), which inhibit LspA, are used in bacterial community internecine competition. LspA is required for full virulence in Gram-positive bacteria[6]. Since resistance development is less of an issue with virulence factors, they are attractive drug targets[7]. A crystal structure of LspA from the opportunistic human pathogen, *Pseudomonas aeruginosa* (LspPae), in complex with globomycin is available[5].

With a view to exploiting this for species-specific drug discovery, a structure of the enzyme from another pathogen was sought. The multi-drug resistant ESKAPE pathogens are a major threat to public health[8]. Among this group, *Staphylococcus aureus* was identified as a fitting target. Here we report high-resolution crystal structures of LspA from methicillin-resistant *S. aureus* (MRSA) (LspMrs) in complex with globomycin and with myxovirescin A1 (hereafter myxovirescin). The structures confirm a mechanism of action, which involves blocking the catalytic dyad in this aspartyl protease. Interestingly, whilst globomycin approaches from one side of the substrate-binding pocket, myxovirescin does so from the other. Where these chemically and structurally distinct antibiotics overlap in the complex provides a blueprint for a drug development program. In combination with the high-resolution structure of LspA from *P. aeruginosa* (LspPae), these two structures set the stage for a campaign aimed at species-specific as well as broad-spectrum drug discovery.

## Results

**Survival of the *S. aureus* *lspA* mutant in human blood.** While *lspA* is essential in Gram-negative bacteria such as *E. coli* and *P. aeruginosa*, the *lspA* gene is not essential in Gram-positive monoderm bacteria including *S. aureus*[6,9]. Previously, LspA-deficient mutants of a methicillin-sensitive *S. aureus* (MSSA) strain were shown to be attenuated in a mouse model of infection and a signature-tagged mutagenesis screen in an MSSA background showed that disruption of *lspA* led to a loss of virulence in a mouse model of bacteremia[10,11]. Here we generated an *lspA* mutant of the community associated MRSA strain LAC. The *lspA* mutant grew similarly to wild-type LAC in rich laboratory media, showing that loss of LspA activity does not affect bacterial growth in vitro (Supplementary Fig. 1a). To determine if LspA activity is important for the survival of MRSA during human infection, ex vivo infection studies were performed with whole human blood. The *lspA* mutant had a reduced ability to survive in whole

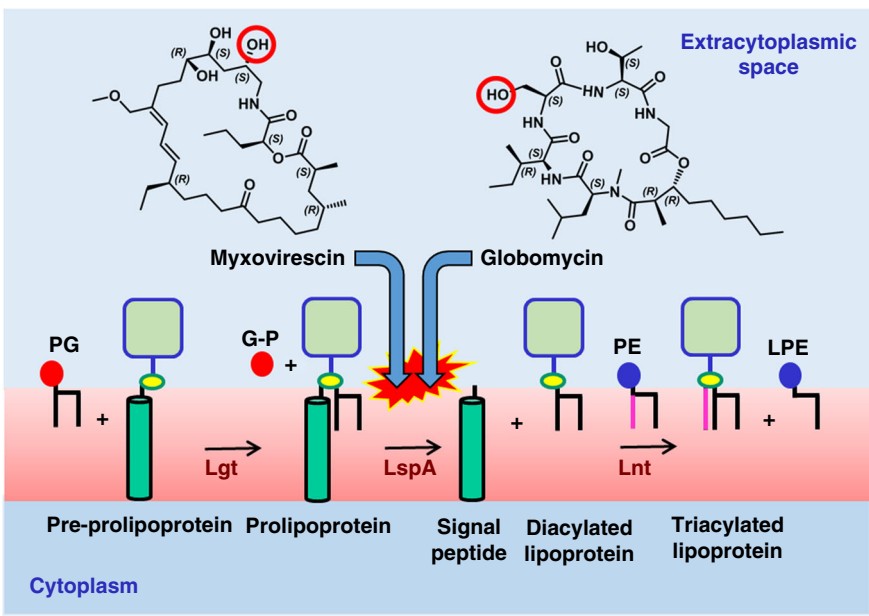

**Fig. 1 Myxovirescin and globomycin target the lipoprotein posttranslational processing pathway.** The pathway in Gram-negative bacteria employs the prolipoprotein diacylglyceryl transferase, Lgt, to dagylate the pre-prolipoprotein, LspA to cleave off the signal peptide from the proliprotein and the apolipoprotein N-acyltransferase, Lnt, to perform a final N-acylation step. LspA is inhibited by both antibiotics. Commonly used lipid substrates include phosphatidylglycerol (PG) and phosphatidylethanolamine (PE). Non-protein products are glycerol-1-phosphate (G-P) and lyso-PE (LPE). Gram-positive bacteria do not have the canonical *Lnt*. However, an Lnt-like activity in Gram-positives has recently been identified[50]. The blocking hydroxyl groups common to both antibiotics that sit between the catalytic aspartates, thereby inhibiting LspA, are identified with red circles.

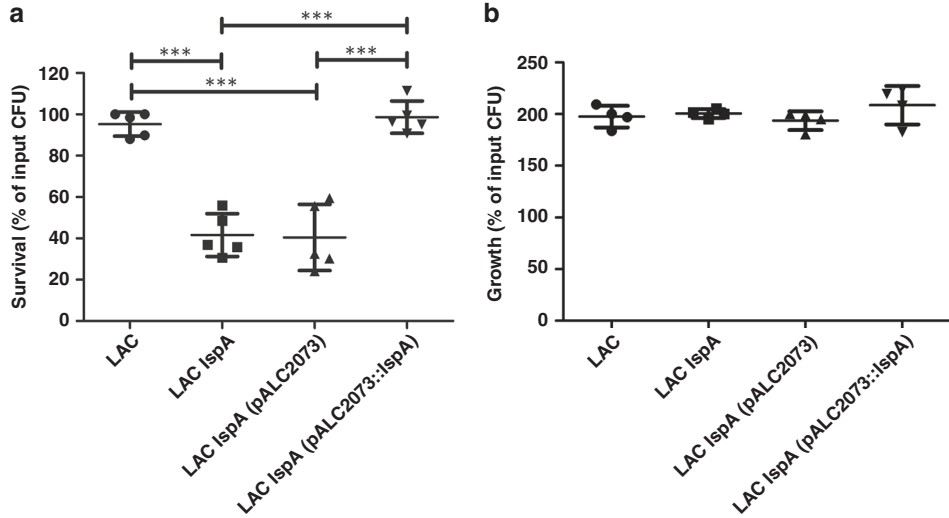

**Fig. 2 Survival of the *S. aureus lspA* mutant in human blood and plasma. a** *S. aureus* LAC and the *lspA* mutant were incubated in whole blood and **b** plasma from healthy human blood donors. LAC corresponds to wild-type *S. aureus*, LAC *lspA* to the inactive mutant, LAC *lspA* (pALC2073) to the mutant carrying the empty multicopy plasmid and LAC *lspA* (pALC2073::*lspA*) to the mutant carrying the multicopy plasmid containing the *lspA* gene. Total viable counts were determined for the initial inoculum ($T_0$) and blood and plasma samples 3 h post-inoculation ($T_3$). The percentage of bacterial survival was determined by dividing the $T_3$ count × 100 by the $T_0$ count. Each data point represents a different blood donor. Statistical significance was determined using one-way ANOVA + Bonferroni's Multiple Comparison Test. \*\*\*$P < 0.0001$. Source data are provided as a Source Data file.

**Table 1 Effect of antibiotics on peptidase activity of LspA constructs determined by gel-shift assay.**

| Construct | Relative activity (%)[a] | IC$_{50}$ (μM)[c,d,e] | |
|---|---|---|---|
| | | Globomycin | Myxovirescin |
| LspPae WT | 133[b] | 0.64 ± 0.16 | 1.09 ± 0.34 |
| LspMrs WT | 100 | 170.7 ± 0.64 | 0.16 ± 0.00 |
| N52A | 7 | 1.23 ± 0.59 | – [f] |
| N52Q | 71 | 2.43 ± 0.34 | 0.29 ± 0.08 |
| G54A | 113 | 1.78 ± 0.25 | 0.57 ± 0.01 |
| G54P | 0 | ND | ND |
| R110A | 0 | ND | ND |
| R110K | 50 | 1.73 ± 0.10 | 0.40 ± 0.03 |
| D118N | 0 | ND | ND |
| N133A | 3 | ND | ND |
| N133Q | 3 | ND | ND |
| D136N | 0 | ND | ND |

Source data are provided as a Source Data file
[a]The values shown are based on product formed after 30 minutes of reaction
[b]Under initial rate conditions (the first 5 min of reaction), the relative activity value is closer to 300% (see Supplementary Fig. 8). After 30 min of reaction, the relative activity is 133%
[c]IC$_{50}$ values are shown as the average (±standard deviation) of duplicate measurements
[d]For reference purposes, minimum inhibitory concentration (MIC) values for globomycin acting on *P. aeruginosa* and *S. aureus* were >100 μg/mL (152 μM)[51]. The most potent globomycin analogue included in that study had corresponding MIC values of >100 μg/mL (152 μM) and 6.3–12.5 μg/mL (9.6–19.1 μM)[51]
[e]MIC values for myxovirescin A acting on *E. coli* has been reported at 1 μg/mL (1.6 μM)[39]
[f]The signal was too weak for quantification using Image Lab software
ND not determined

human blood compared to the wild-type (Fig. 2a). The ability of LAC *lspA* to survive in blood was restored by complementation with the *lspA* gene on a multi-copy plasmid (LAC *lspA* (pALC2073::*lspA*)). These results show that LspA activity is important for the survival of MRSA in human blood. As a control, the same strains and mutants were incubated in fresh plasma from the same blood donors. There was no reduction in the ability of the *lspA* mutant to grow in human plasma (Fig. 2b) under the same conditions. This indicates that the *lspA* mutant has a reduced ability to survive killing by phagocytes in human blood rather than having a growth defect under the conditions

used. These findings identified the LspA enzyme as an attractive target for treatment of MRSA infection. In support of a structure-based drug design campaign we next sought to determine a high-resolution X-ray crystal structure of LspA from MRSA (LspMrs).

**LspMrs-globomycin complex structure.** Recombinant LspMrs was produced and purified using in vivo methods of expression (Supplementary Fig. 2). The enzyme is 163 residues long and was hexahistidine-tagged. It was shown to be active using a gel-shift assay with a recombinant prolipoprotein, inhibitor of cysteine protease (proICP), as substrate (Table 1, Supplementary Figs. 3a and 4, Supplementary Table 2). LspMrs was also assayed by fluorescence resonance energy transfer (FRET) using a single molecule FRET lipopeptide (Supplementary Figs. 3b and 5, Supplementary Table 2). Under optimized conditions, the enzyme, at a concentration of 0.3 μM, had an estimated apparent $K_m$ of 47 μM and $V_{max}$ of 2.5 nmol/(mg min), as determined by FRET assay. The corresponding values for LspPae were 10 μM and 107 nmol/(mg min) at an enzyme concentration of 0.1 μM, suggesting that the *S. aureus* ortholog is a slower acting peptidase with a lower substrate affinity than its *P. aeruginosa* counterpart (see Supplementary Discussion). The purified enzyme from both organisms was inhibited by globomycin with IC$_{50}$ values approaching the enzyme concentration used for assay (Supplementary Fig. 6, Supplementary Table 3). This is consistent with tight binding inhibition[12,13] (Supplementary Discussion). When proICP was used as substrate, the IC$_{50}$ of LspPae for globomycin was 0.64 μM at an enzyme concentration of 0.5 μM, again consistent with tight binding. However, for LspMrs the value was 171 μM at an enzyme concentration of 0.5 μM (Supplementary Fig. 4, Table 1). The latter insensitivity to globomycin with a lipoprotein substrate may reflect the ortholog's sequence and structure. It may also have to do with the substrate identity and concentration used in the assay, as discussed (Supplementary Discussion).

Initial attempts at crystallizing recombinant, hexahistidine-tagged LspMrs in its apo- and globomycin bound forms using the in meso method, as implemented with LspPae, were not successful. However, crystals and a structure to 1.92 Å resolution

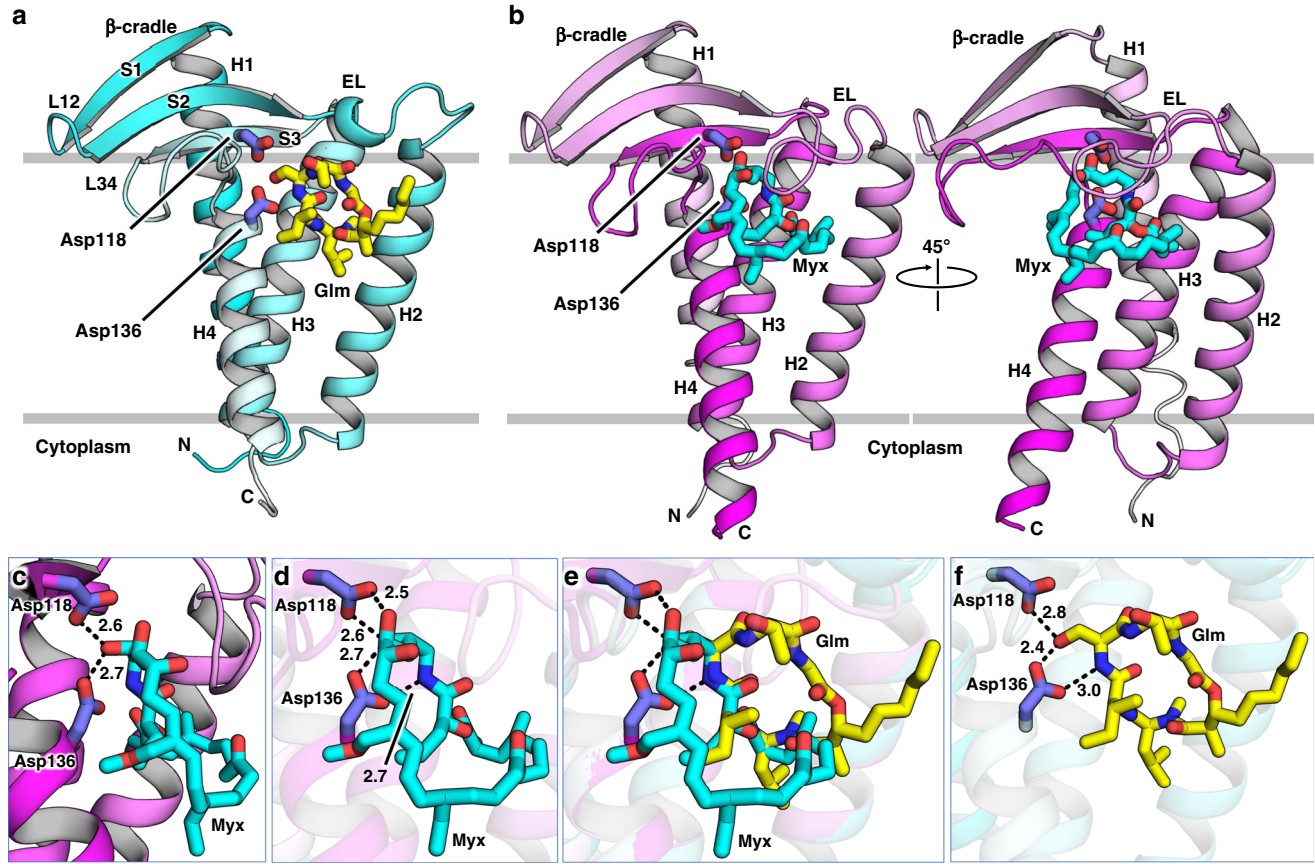

**Fig. 3 Structures of LspMrs in complex with globomycin and with myxovirescin. a, b** Views from the membrane into the binding pocket containing globomycin and myxovirescin, respectively. **c** Detailed view of the blocking hydroxy group (6-OH) of myxovirescin interacting with the catalytic aspartate residues. **d–f** Comparison of myxovirescin and globomycin in the binding pocket of LspMrs. Superposed protein structures are shown in the background with parts removed for clarity. Catalytic aspartate dyad residues are shown with carbons in dark blue. A superpose of the two antibiotics in the binding pocket is shown in **e**. Because myxovirescin has a complicated bent ovoid shape (Supplementary Fig. 11) it is impossible to find a single orientation where all of its non-hydrogen atoms are visible. For a fuller appreciation of myxovirescin's molecular shape, the reader is referred to the Protein Data Bank record (PDB ID, 6RYP). In **b**, the C-terminal residues $K^{157}$, $K^{158}$, $E^{159}$, $K^{160}$, $E^{161}$, $V^{162}$ and $K^{163}$, which extend into the cytoplasm as a helix/coil, have been deleted for a more uniform comparison with **a** where these residues are disordered. Distances are shown in ångströms as dashed lines.

were obtained for the LspMrs-globomycin complex when the tag was removed (Fig. 3a and Supplementary Fig. 2, Table 2). With a sequence identity between LspMrs and LspPae of 31%, it was not surprising that the structures of the two complexes were similar with a Cα r.m.s.d. of 0.713 Å over 155 residues. The enzyme consists of four transmembrane helices (H1-H4) with the catalytic dyad aspartates, Asp118 and Asp136, towards the membrane's outer surface. Extending away from the catalytic dyad and to one side of the bundle of transmembrane helices is the β-cradle, a hemi-cylindrically shaped sheet that sits on the membrane. It is proposed to accommodate the stretch of residues to the C-side of the LspA cleavage site in lipoprotein substrates. An extracellular loop (EL) between strand 2 (S2) in the β-cradle and H2 sits above the substrate binding pocket in which resides globomycin. Globomycin, a 19-member cyclic depsipeptide that includes five amino acids (g.Ser, g.Thr, g.Gly, g.Leu g.Ile; the g refers to globomycin) and an α-methyl-β-hydroxy nonanoate (Fig. 1 and Supplementary Fig. 3c), inhibits LspA by lodging the β-hydroxyl of its g.Ser between the catalytic aspartates. Our interpretation is that the antibiotic acts as a non-cleavable tetrahedral intermediate analog as has been proposed for many inhibitors of other aspartyl proteases[14]. The 14 highly conserved residues (Asp21, Lys25, Asn52, Gly54, Gly102, Asn106, Asp109, Arg110, Val116, Asp118, Phe132, Asn133, Ala135, Asp136) in LspMrs cluster around globomycin in the active and

substrate binding sites in ways that make good biochemical sense (Supplementary Fig. 7), as described for LspPae[5]. Site-directed mutagenesis supports the view that the catalytic dyad residues are Asp118 and Asp136 (Table 1). Contacts between globomycin in LspMrs and in LspPae are very similar (Supplementary Table 4)[5].

The prolipoprotein substrate of LspA consists of an N-terminal signal peptide of about twenty amino acids followed by the consensus lipobox $[L(VI)^{-3}-A(STVI)^{-2}-G(AS)^{-1}-C^{*+1}]^{15}$ and the functional C-terminal protein domain (Supplementary Fig. 3a). LspA cleaves between the G and the C* in the lipobox where C* represents a diacylglyceryl (DAG)-modified (dagy-lated) cysteine. The signal peptide is assumed to be helical and to be accommodated in the membrane in the space created by the orthogonally packed H2, H3 and H4 in LspA (Supplementary Fig. 9). The lipobox is proposed to take the form of an extended peptide with the scissile bond between Gly and Cys* next to the catalytic dyad aspartates. It is the lipobox sequence that the g.Leu-g.Ile-g.Ser in globomycin is assumed to mimic. The acyl chains of the DAG in C* have been modeled to situate below the β-cradle (Supplementary Fig. 9). Evidence in support of this model comes from the location of monoolein lipid molecules in the crystal structure that derive from the cubic mesophase in which crystallization was performed (Supplementary Fig. 10)[5].

**Table 2 Data collection and refinement statistics[a].**

| | LspMrs-globomycin (6RYO) | LspMrs-myxovirescin (6RYP) |
|---|---|---|
| Data collection | | |
| Space group | $P3_221$ | $P6_122$ |
| Cell dimensions | | |
| $a, b, c$ (Å) | 52.3, 52.3, 135.9 | 54.2, 54.2, 317.5 |
| $\alpha, \beta, \gamma$ (°) | 90, 90, 120 | 90, 90, 120 |
| Beamline | SLS-X06SA | SLS-X06SA |
| Wavelength (Å) | 0.92003 | 1.0 |
| Resolution (Å) | 45.26–1.92 (2.23–1.92)[b] | 46.40–2.30 (2.36–2.30) |
| No. of reflections (total/unique) | 33,655/7,051 | 199,005/23,038 |
| $R_{meas}$ | 0.17 (1.38) | 0.14 (2.79) |
| $I/\sigma_I$ | 7.00 (1.80) | 7.69 (0.90) |
| Completeness (%) | 86.20 (62.50)[c] | 99.9 (100) |
| Multiplicity | 4.80 (3.70) | 8.63 (8.90) |
| $CC_{1/2}$ | 0.99 (0.47) | 0.99 (0.22) |
| Refinement | | |
| Resolution (Å) | 45.26–1.92 | 46.40–2.30 |
| Reflections | 7046 | 23,038 |
| $R_{work}/R_{free}$ | 0.25/0.28 | 0.25/0.28 |
| R.m.s. deviations | | |
| Bond lengths (Å) | 0.002 | 0.001 |
| Bond angles (°) | 0.498 | 0.372 |
| No. of atoms | | |
| Protein | 1232 | 1287 |
| Globomycin or Myxovirescin | 46 | 44 |
| Other ligands[d] | 242 | 176 |
| Water | 28 | 12 |
| B-factor | | |
| Protein | 36.78 | 68.15 |
| Globomycin or Myxovirescin | 20.87 | 75.61 |
| Other ligands[d] | 58.91 | 100.05 |
| Water | 29.36 | 75.49 |
| Ramachandran Plot | | |
| Favored (%) | 97.39 | 98.15 |
| Allowed (%) | 2.61 | 1.85 |
| Outliers (%) | 0 | 0 |
| MolProbity Clashscore | 7.87 | 1.32 |

[a]Data processing statistics are reported with Friedel pairs merged for 6RYO and Friedel pairs separated for 6RYP
[b]Values in parentheses are for the highest resolution shell
[c]Ellipsoidal completeness, as defined by *autoPROC/STARANISO*, is referred to here
[d]Other 'ligands' include monoolein, glycerol, PEG, $SO_4^{2-}$ and $Zn^{2+}$

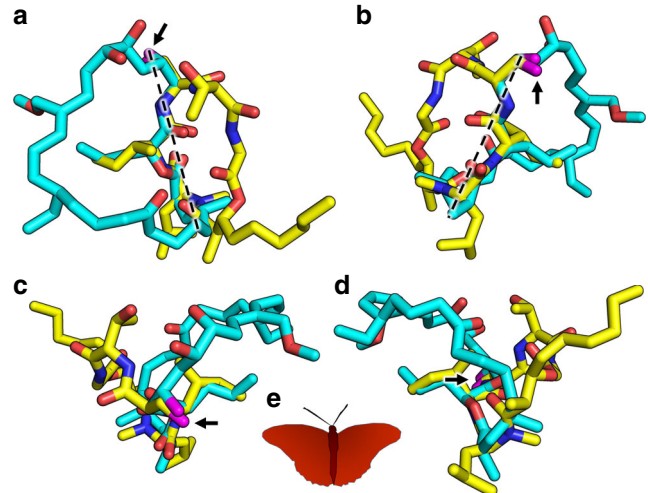

**Fig. 4 Overlay of myxovirescin and globomycin in the LspMrs complex structures.** The superpose identifies atom positions along the 'spine' that are common to the two antibiotics. **a** View from the membrane into the active site. **b** View from the active site. **c** View along the spine (bottom) from the extracellular space. **d** View along the spine (bottom) from the cytoplasm. Myxovirescin and globomycin are shown with carbon atoms colored cyan and yellow, respectively. The blocking hydroxyl in myxovirescin and globomycin is shown in magenta and is marked with a black arrow. A dashed line in **a**, **b** is included to guide the eye in locating the spine feature. **e** With a common spine and the macrocycles splayed apart, the image of the two superposed antibiotics gives the impression of a butterfly with outstretched wings.

Together, these structural and biochemical data show that LspA from MRSA, a Gram-positive bacterium, is a close ortholog of its counterpart in the Gram-negative *P. aeruginosa*. One noted difference however is in the sensitivity of LspA in the two organisms to globomycin when assayed by the gel-shift method with proICP as substrate (Table 1).

**LspMrs-myxovirescin complex structure.** Myxovirescin, also known as antibiotic TA, megovalicin or M-230B, is a 28-membered macrolactam lactone[16] (Fig. 1 and Supplementary Fig. 3c). It is an inhibitor of LspA and a potential broad-spectrum antibiotic with desirable adhesive properties. Myxovirescin has been used in clinical trials for the treatment of plaque and gingivitis[17] and has been proposed for use in limiting infection rate associated with indwelling catheters[18]. Despite both myxovirescin and globomycin being LspA inhibitors, their chemical makeup (Fig. 1) and the pathways used for in vivo synthesis are entirely different. Myxovirescin is a secondary metabolite produced by *Myxococcus xanthus* while globomycin is a product of *Streptomyces hagronensis*[19,20]. Other than being cyclic and containing hydroxyl, ester and amide moieties the two antibiotics are chemically distinct. Indeed, attempts 'by eye' failed to predict the binding pose of myxovirescin observed in the final crystal structure based on that for the LspMrs-globomycin complex

(Fig. 3a). In FRET and gel-shift assays, myxovirescin was shown to be a potent inhibitor of LspMrs and LpsPae (Supplementary Figs. 4 and 6, Table 1).

Despite extensive efforts, attempts at generating a crystal structure of myxovirescin in complex with LspPae failed. However, when combined with LspMrs, crystals and a structure of the complex to 2.30 Å resolution were obtained by the in meso method using monoolein as the host lipid (Fig. 3b and Table 2). Overall, the structure of the protein in the myxovirescin complex is very similar to that of the complex with globomycin. Differences were noted at the C-terminus that was fully formed as an extended helix in the myxovirescin complex but was disordered, and without useful electron density in the globomycin complex. Another important difference at the EL is described below.

**Differences and similarities in the complex structures.** What came as a surprise when the structures of LspMrs in complex with globomycin and myxovirescin were compared was the finding that the two antibiotics interact with the enzyme by associating with opposing sides of the substrate-binding pocket (Fig. 3d–f). However, at the catalytic dyad, which presumably is where it matters, both engage identically. Looking into the binding pocket from the membrane plane, globomycin binds with its g.Ser hydroxyl lodged between the catalytic dyad aspartates and with the rest of the depsipeptide ring bound to the right side of the pocket neatly tucked in under a retracted EL (Fig. 3a). By contrast, myxovirescin binds with its macrocycle docked on the left side of the pocket with the EL extended out and over the ring (Fig. 3b). However, despite this marked difference in docking mode and approach to the active site, the 6-OH of myxovirescin wedges firmly between the two catalytic aspartates inhibiting the enzyme (Fig. 3c), just like globomycin, as a non-cleavable

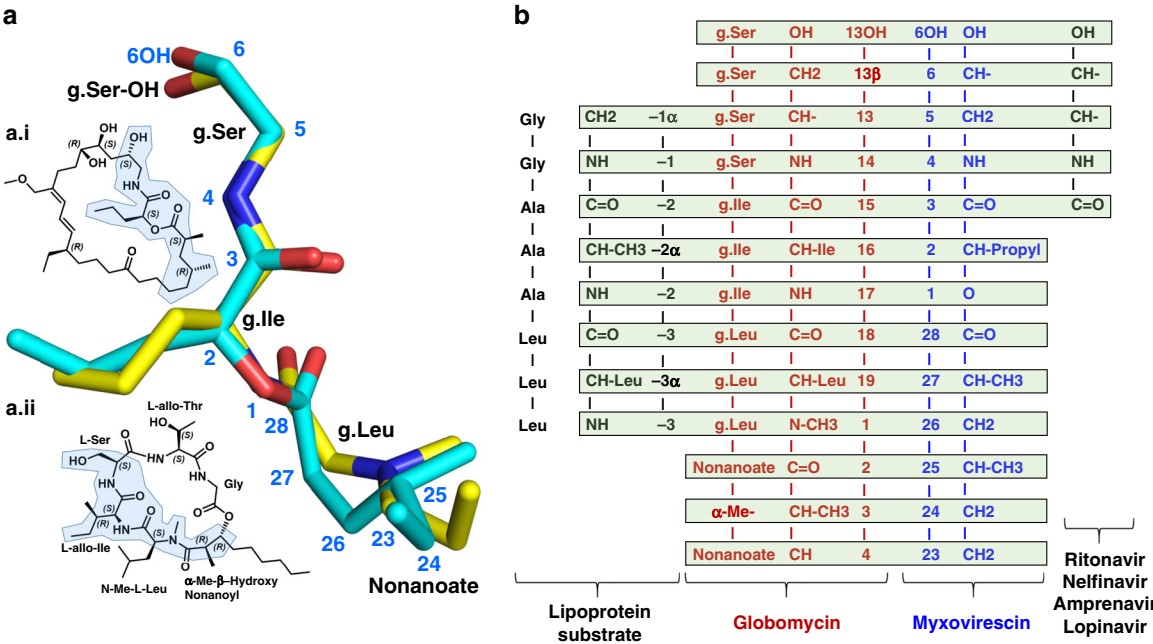

**Fig. 5 Atoms in globomycin and myxovirescin with common positions in the complex structures.** These are referred to in the text as 'spine atoms'. **a** Superpose of the atoms in globomycin (yellow carbons, black labels) and myxovirescin (cyan carbons, blue labels) from crystal structures of the corresponding LspMrs complexes. Blocking hydroxyls are at the top of the image; the two oxygens are 0.8 Å apart. Nineteen atoms along the spine of the two antibiotics are within 1.3 Å of one another. (a.i). Chemical structure of myxovirescin. (a.ii). Chemical structure of globomycin. Overlapping atoms in the corresponding structures are shaded. **b** Alignment of the sequence of atoms and chemical groups in globomycin and myxovirescin that overlap in the structures of the corresponding LspMrs complexes. To the left and right, respectively, are shown parts of the consensus lipobox sequence (-LAGC-) in a prolipoprotein substrate and a non-cleavable tetrahedral intermediate analog isostere found in human immunodeficiency virus (HIV) protease inhibitor drugs as they might appear in complex with LspA[14].

tetrahedral intermediate analog. Indeed, the 6-OH of myxovirescin and the β-OH of g.Ser, hereafter referred to as blocking hydroxyls, superpose to within 0.8 Å in the two complex structures (Fig. 3a, c). Given the disparate natures and biosynthetic origins of the two antibiotics, this agreement in the placement of the blocking hydroxyls is remarkable.

The similarities do not end there. Tracking along the myxovirescin molecule from the 6-OH to the [-4NH-3C(O)-] amide linkage (Supplementary Fig. 3c) we find the latter hydrogen bonded to the catalytic Asp136 and to Arg110. The exact same interaction exists between the g.Ser-to-g.Ile amide linkage in globomycin and Asp136 and Arg110 in LspMrs (Supplementary Table 4). Indeed, the sequence of six consecutive atoms from 6-OH to 3-C(O) in the myxovirescin complex superpose with identical atoms in the globomycin complex (Fig. 4). This impressive overlap extends for another thirteen contiguous atoms to 23-C in myxovirescin which comes close to aligning with 4-C in globomycin. The correspondence of all nineteen, what we refer to as spine atoms, is shown in Fig. 5. Beyond the spine termini at 6-C and 23-C in myxovirescin and 4-C and 13-C in globomycin, the two rings diverge and splay apart with the myxovirescin ring oriented to the left and the globomycin ring to the right of the substrate binding pocket (Figs. 3d, f and 4a).

It is interesting to see globomycin partition to one half of the substrate-binding pocket while myxovirescin partitions to the other. Both macrocycles have polar and apolar sides. For the most part, it is the former that overlap along the spines in the complex structures (Fig. 5). The apolar parts extend away from the spine and to the left and right halves of the substrate-binding pocket (Fig. 4a). In myxovirescin, the apolar part situates partly below the overarching β-cradle. This is reminiscent of the proposed

binding mode for the acyl chains of the DAG in the prolipoprotein substrate (Supplementary Fig. 9). In the case of globomycin, the g.Ile and g.Leu apolar side chains and the nonanoate acyl chain extend from the bottom of the ring. The latter wraps around the membrane embedded portion of H2 below the apolar surface of the EL. Thus, the two antibiotics appear to have evolved to mimic different features of the natural substrate for better association with the substrate binding surfaces of the enzyme to properly position their blocking hydroxyls between the catalytic dyad residues.

With a common spine of nineteen atoms and the macrocycles splayed apart as described, the image of the two superposed antibiotics gives the impression of a butterfly with outstretched wings (Fig. 4e). The body of the butterfly is the spine of atoms 'shared' by both antibiotics; the two divergent macrocycles and blocking hydroxyls correspond to the butterfly wings and head, respectively. Relatedly, of note is the similarity between the 'business end' of the spines that engages with the catalytic dyad and the hydroxy-ethylamine isostere developed as the active pharmacophore in several FDA approved HIV-1 (aspartyl) protease inhibitors (Fig. 5b)[14].

The plane of the globomycin ring is relatively flat. By contrast, in complex with LspMrs, myxovirescin assumes the shape of an ovaloid disc bent at its broad end (Figs. 3 and 4, Supplementary Fig. 11). The flat face and bent portions of the disc include ring members 1–16 and 17–28, respectively. The former faces the active site positioning the blocking 6-OH between the catalytic dyad residues. The latter extends orthogonally away from the surface of the substrate-binding pocket and toward the apolar recesses of the membrane. Interestingly, the overlap between globomycin and myxovirescin includes both upright and bent sides of the macrolactam ring (Fig. 4). Equally interesting is the

overlap between the apolar side chain of g.Ile and the propyl moiety of myxovirescin. In the former, the chain points out of the globomycin ring, in the latter it points into the macrolactam ring (Fig. 4a and supplementary fig. 3c). The symmetry in the superposed antibiotic structures extends to the long acyl chain in globomycin which is matched by the apolar sequence of atoms (ring atoms 12–23) in myxovirescin to which is attached an outstretched ethyl group (Supplementary Fig. 3c). The impression given by these apolar regions is that they contribute to stabilizing the interaction between the antibiotics and the enzyme through shape and hydrophobic complementarity with different membrane integral parts of the protein.

All fourteen of the highly conserved residues in LspA are similarly poised in the globomycin and myxovirescin complex structures, with the exception of Gly54 which facilitates the flexibility of the extra-cytoplasmic loop, as discussed below. Half of these are in hydrogen bonding distance to either antibiotic (Supplementary Fig. 12, Supplementary Table 4) along the ring spine referred to above. One of the conserved residues, Asn52, is particularly interesting. Site-directed mutagenesis attest to its importance in LspA function (Table 1). It interacts with both antibiotics where they have just splayed apart in the superpose at the extra-membrane entrance to the active site (Supplementary Fig. 13). Its Nδ2 forms a hydrogen bond with the 8-OH in myxovirescin and with the γ-OH on g.Thr in globomycin. Asn52 is part of the β-cradle and is held in place by hydrogen bonds to the backbone amide and carbonyls of highly conserved Val116 which also is in the β-cradle. It seems remarkable that these two entirely different antibiotics have evolved to exploit for binding in very different ways the same conserved residues in LspA presumably to the same ends of tightly binding to and inactivating the enzyme.

**EL flexibility**. It was noted that an important difference between the structures of LspMrs with myxovirescin and globomycin was at the EL, a sequence of 11 residues from Asn53 to Lys63. In the globomycin complex, the loop includes a half-turn helix from which extends conserved Trp57. The tryptophan reaches over the globomycin molecule and secures it in place against one side of the substrate-binding surface of the enzyme by hydrogen bonding. In the myxovirescin complex, the loop has unfolded fully and extends deeper into the substrate-binding pocket enabling the benzene ring of Trp57 to contact the macrocycle locking it in position against the opposite side of the pocket surface (Supplementary Fig. 14). Thus, EL flexibility is key to effective globomycin and myxovirescin binding to LspA. It seems unlikely that LspA evolved this loop pliability unless it furthered its own ends, one of which is to process lipoproteins. Given the considerable number of substrate lipoproteins LspA must process (175 in Pae, 67 in Mrs), EL flexibility may well reflect a certain level of necessary substrate promiscuity. Gly54 is the second residue in the loop and is expected to play a role in flexibility. Indeed, mutating it to Pro, a residue known to limit mobility, fully inactivates the enzyme with lipoprotein substrate (Table 1). Together these data highlight how globomycin and myxovirescin have exploited functional loop flexibility to their respective ends through convergent evolution.

## Discussion
The fact that both globomycin and myxovirescin interact closely with many of the highly conserved residues in LspA (Supplementary Figs. 7 and 12) raises an interesting point concerning their resistance hardiness. Mutations in these residues in isolation that weaken the enzyme's interaction with the antibiotics might be expected to reduce antibacterial potency. At the same time however, mutating conserved residues will compromise the host

bacterium as a result of inactivating, destabilizing or attenuating expression of LspA. No competitive advantage would be gained by doing so. It appears therefore that globomycin and myxovirescin may have evolved to be 'resistance-proof' and, in principle, should be attractive as therapeutic agents or leads. It can be difficult and in some cases impossible to test this hypothesis of resistance hardiness by means of site-directed mutagenesis where both potency and affinity are monitored based on enzyme activity. However, a number of such mutants have been prepared and their peptidase and antibiotic sensitivities have been measured (Table 1). Interestingly, in all cases the $IC_{50}$ for globomycin determined using a lipoprotein substrate dropped compared to wild-type values. It would appear that other interactions are called into play to compensate for the changes introduced in the mutant constructs that make it an even more potent inhibitor. Crystal structures of these mutants in complex with globomycin would shed light on this proposal and may prove useful in drug design. Nonetheless, it makes sense that lessons in resistance hardiness and longevity provided by nature should be applied to structure-based drug discovery strategies. Specifically, this would take the form of incorporating into the design of therapeutics interactions between the lead compound and conserved residues in the target protein. Useful information in support of drug design might also derive from a study of the LspA enzymes in the organisms that produce globomycin and myxovirescin.

Given that the organisms responsible for producing globomycin and myxovirescin come from different lineages of bacteria, the two antibiotics provide a convincing example of convergent evolution. Globomycin is produced by at least five *Streptomyces* species[20] belonging to the phylum actinobacteria. *Streptomyces* are aerobic, Gram-positive, immotile filamentous bacteria containing branching mycelia and aerial hyphae where chains of spores form in a manner that resembles most filamentous fungi[21]. By contrast, *Myxococcus xanthus*, the source organism for myxovirescin, belongs to the phylum proteobacteria, and is a Gram-negative bacteria, known for its social behaviour, gliding motility, and fruiting body and myxospore formation[22]. Both bacterial species are found in soil, a microbe-rich environment, where they use the products of secondary metabolism for defence. Otherwise, these two antibiotic producing microorganisms have little in common. Indeed, they are as far apart in the kingdom bacteria as can be. All bacterial species contain the *lspA* gene for purposes of primary metabolism. However, there are only two bacterial genera, *Streptomyces* and *Myxococcus*, known to produce molecules that target LspA. There is no evidence of an ancestral molecule targeting the LspA active site.

Convergent evolution refers to the acquisition over time of similar form and function in organisms of distinct genetic origin. In the current example, that commonality finds expression in two entirely unrelated secondary metabolites. These have been crafted through the action of profoundly different multistep pathways in different bacterial species to target identically a specific catalytic dyad in the lipoprotein processing enzyme, LspA. Myxovirescin is synthesized from short chain carboxylates by a polyketide synthase[19]. By contrast, a giant nonribosomal peptide synthetase is likely implicated in globomycin synthesis. Remarkably, the two antibiotics have achieved potency and specificity through a critical hydroxyl group that extends from profoundly different molecular scaffolds (Fig. 1). In myxovirescin, the scaffold is a macrolactam lactone. In globomycin, it is a cyclic depsipeptide. Therefore, the biosynthetic machineries in *Myxococcus* and *Streptomyces* have been able to fabricate molecular features in myxovirescin and globomycin, respectively, to mimic similar as well as distinct characteristics of the natural prolipoprotein substrate and the tetrahedral transition state intermediate it presumably forms during proteolytic cleavage.

Other examples of natural antibiotics that may have emerged through convergent evolution include rifampicin and sorangicin, which target the β subunit pocket in RNA polymerase (RNAP)[23] and ripostatin and corallopyronin which bind to the switch region of RNAP[24].

As noted, nature has evolved globomycin and myxovirescin to target lipoprotein posttranslational processing at the peptidolytic step catalysed by LspA. The active site in the enzyme, consisting of catalytic dyad aspartates, is specifically targeted by both. Each includes a hydroxyl group suitably disposed for placement between the dyad residues and mimicking, in a non-cleavable analog, the hydroxyl of the putative tetrahedral intermediate. In the structures of LspA complexed with globomycin and with myxovirescin these blocking hydroxyl oxygens superpose to within 0.8 Å of one another (Fig. 5). This precise placement undoubtedly reflects a multitude of molecular features in the two antibiotics. These include a common spine of 19 contiguous atoms and surrounding macrocycles that target, respectively, the proposed signal peptide and diacylglyceryl binding surfaces in LspA.

Clearly, the 19-atom long spines in globomycin and myxovirescin are proven and effective and can be viewed as natural pharmacaphores. Their chemical composition and structure can be used to inform drug design and discovery efforts with a view to producing more potent broad spectrum as well as species-specific antibiotics. In the case of globomycin and myxovirescin, both are macrocycles and it may be that this feature must be retained in related newly produced therapeutics. However, the scaffold used to make the ring can be optimized separately for pharmacokinetic properties and for outer membrane permeability when targeting Gram-negative pathogens. Provided the synthesis can be realized and the product is not too bulky, it might be worth generating a bicyclic compound inspired by the superposition of the two antibiotics in the complex structures. One ring in the bicycle would mimic that in globomycin, the other the scaffold in myxovirescin (related to our suggestion to use the binding poses of globomycin and myxovirescin in LspA to inform the generation of a more effective bicyclic compound, the recently discovered natural antibiotic darobactin is bicyclic, consisting of two fused peptide rings[25]). This would then combine in the one new chemical entity features that target simultaneously the active site as well as the proposed signal sequence and diacylglyceryl binding surfaces of the substrate-binding pocket delivering high potency and specificity. A long-term goal of the current project is to explore these options along with structures of complexes with orthologs from pathogenic and benign species to aid in tailoring antibiotics that are either broad spectrum or species-specific in their application.

We have shown that LspA is an ideal target for anti-infective agents as it is required for the survival of MRSA under physiologically relevant conditions in human blood. Thus even in bacterial species where LspA is not essential there is potential for therapeutic benefit using drugs that target LspA.

## Methods

**Bacterial strains and mutants**. The *S. aureus* LAC strain used here is an erythromycin sensitive derivative of the USA300 MRSA strain LAC (a gift from A. Horswill, University of Colorado)[26]. An *lspA*-deficient mutant of LAC was generated by phage transduction of the *lspA::*Tn cassette from the NEBRASKA library JE2 strain[27] using φ85. Resulting transductants were selected on erythromycin-containing agar and the presence of the transposon was confirmed by carrying out the polymerase chain reaction on genomic DNA using primers that anneal upstream and downstream of the *lspA* gene and by DNA sequencing of the polymerase chain reaction product. LAC *lspA* was phenotypically indistinguishable from LAC in terms of growth profile in tryptic soy broth (TSB) (Supplementary Fig. 1a) and haemolysis patterns on sheep blood agar (Supplementary Fig. 1b).

Complementation of the *lspA* mutant was achieved by cloning the *lspA* gene from LAC into the expression plasmid pALC2073[28] between EcoRI and KpnI

restriction sites to generate pALC2073::*lspA*. The *lspA* gene was amplified using the LspA Forward (5′-GCAGGTACCATTGGAGGAACGAAAATGCAC-3′) and LspA Reverse (Lsp, 5′-GCCATGAATTCCTCCATTACTTAACCTCCTTCTCC-3′) primers. The mutant was transformed with plasmid DNA using electroporation. Empty plasmid pALC2073 was introduced into the mutant to serve as a control.

**Survival and growth in human blood and plasma**. Human blood was obtained from healthy volunteers between 20 and 40 years of age. Ethical approval for the use of human blood was obtained from the Trinity College Dublin Faculty of Health Sciences ethics committee.

Freshly drawn blood was collected in tubes containing the anti-coagulant hirudin (Refludan, Pharmion). The plasma fraction was obtained by centrifugation of the blood at $4,000 \times g$ for 10 min. To quantify survival of *S. aureus* in human blood and plasma the method described by O'Halloran et al.[29] was followed. Briefly, *S. aureus* was grown in TSB overnight at 37 °C for 16–18 h. The bacteria were diluted into RPMI-1640 medium to a final density of *ca.* $1 \times 10^4$ CFU/mL and 25 μL of this was added to 475 μL of blood or plasma. Tubes were incubated at 37 °C with gentle rocking, and after 3 h serial dilutions were plated in triplicate to determine the CFU/mL of viable bacteria and the percentage survival of the original inoculum was determined.

**Expression and purification for structure determination**. The recombinant construct used in this study was the full length, wild-type LspA from methicillin resistant *Staphylococcus aureus* (strain MRSA252, LspMrs). The sequence of LspA from MRSA252 differs from the LAC sequence at residue 111 only (valine in MRSA252, isoleucine in LAC). It consisted of an N-terminal MG sequence followed by a hexahistidine-tag, the tobacco etch virus (TEV) protease spacer sequence DYDIPTT, the TEV protease cleavage site ENLYFQ/G, an AH linker sequence and the *lspA* gene. The gene, codon optimised for *E. coli* expression using the OptimumGene™ program provided on the GenScript web server (www.genscript.com), was commercially synthesised (GenScript) and sub-cloned into the expression vector pET28a using the restriction sites NdeI and XhoI. The pET28a-*lspA* recombinant plasmid was transformed into chemically competent *E. coli* C43 (DE3) (Lucigen) cells. Cells were grown in TB media supplemented with 50 μg/mL kanamycin (Melford) at 37 °C to an optical density at 600 nm (OD$_{600}$) of 0.5–0.6 and gene expression was induced with 1 mM isopropyl β-D-1-thiogalactopyranoside at 30 °C and 180 rpm for 18 h. Cells were harvested by centrifugation at $6000 \times g$ for 15 min at 4 °C. Cells were either used immediately for purification or stored at −70 °C for a maximum of 2 months.

For purification, cells were re-suspended in 1 g/mL ice-cold Buffer A (50 mM MES/NaOH pH 6.15, 150 mM NaCl, 10%(v/v) glycerol). A single cOmplete™ Protease Inhibitor Cocktail tablet (Roche) was added per 50 mL of re-suspended cells. Cells were lysed by passing three times through a high-pressure homogenizer (Emulsiflex-C5, Avestin®) at 1,500 bar and 4 °C. Cell debris was removed by centrifugation at $25,000 \times g$ for 25 min at 4 °C. The supernatant was ultra-centrifuged at $150,000 \times g$ for 1.5 h at 4 °C. The membrane pellet was re-suspended in 100 mL Buffer A and 1%(w/v) LMNG (Anatrace). Solubilisation was carried out for 18 h at 4 °C on a rotor circulating at 10 rpm. Unsolubilised material was pelleted by ultra-centrifugation at $100,000 \times g$ for 1 h at 4 °C. The supernatant was supplemented with 20 mM imidazole and incubated with 5 mL Ni-NTA Superflow resin (Qiagen) for 1 h at 4 °C on a rotor circulating at 20 rpm for immobilised metal-ion affinity chromatography (IMAC) purification. The resin was transferred to a 100 mL Bio-Rad Econo Column (4 cm diameter) and was washed with 100 mL Buffer B (50 mM MES/NaOH pH 6.15, 300 mM NaCl, 10%(v/v) glycerol, 0.02% (w/v) LMNG, 40 mM imidazole). Bound protein was eluted with 20 mL Buffer C (50 mM MES/NaOH pH 6.15, 300 mM NaCl, 10%(v/v) glycerol, 0.02%(w/v) LMNG, 300 mM imidazole). In all, 1 mL fractions were collected and protein concentration was determined by measuring absorbance at 280 nm with a Nanodrop-1000 microvolume spectrophotometer (Wilmington, DE, USA) using a calculated extinction coefficient of 15,930 M$^{-1}$ cm$^{-1}$ at 280 nm and a molecular weight of 21.2 kDa (ExPASy ProtParam server)[30]. Fractions containing ≥ 0.2 mg/mL LspMrs were combined and desalted using PD10 columns (GE Healthcare). The tobacco etch virus protease (TEVp) was added to the desalted protein solution at a final 1/10 molar ratio of TEVp/LspA. TEVp cleavage was carried out for 18 h at 4 °C. The cleaved and uncleaved LspMrs were separated by passing the sample through 5 mL Ni-NTA Superflow resin. Cleaved LspMrs was collected in the flow-through and concentrated to a volume of 1.5 mL using a 15 mL Millipore centrifuge filter with a molecular weight cut-off of 50 kDa. The concentrated protein sample was loaded onto a HiLoad 16/600 Superdex 200 size-exclusion column pre-equilibrated with 140 mL Buffer D (50 mM MES/NaOH pH 6.15, 150 mM NaCl, 10%(v/v) glycerol, 0.02%(w/v) LMNG) using a GE Healthcare ÄKTA Purifier system. Size-exclusion chromatography was performed at 4 °C at a flow rate of 1 mL/min. Peak fractions with protein concentrations above 0.3 mg/mL were analyzed by SDS-PAGE and fractions containing pure LspMrs were combined and concentrated to 14-16 mg/mL using a Millipore centrifuge filter with a molecular weight cut-off of 50 kDa. The pure protein was divided into 20 μL aliquots, flash frozen in liquid nitrogen and stored at −70 °C.

**Expression and purification for enzyme assays.** The *lspA* gene was sub-cloned into a modified pETDuet-1 vector which facilitated higher expression and purification yields. The recombinant construct was the full length wild-type LspMrs with an N-terminal MGSS sequence followed by a hexahistidine tag, the linker sequence SSGR, the TEV protease cleavage site ENLYFQ/G and a AH linker sequence. Upon cleavage of the N-terminal tag by the TEV protease, the protein sequence was identical to the one used to obtain crystallographic data.

*LspA* mutant genes (N52A, N52Q, G54A, G54P, R110A, R110K, D118N, N133A, N133Q and D136N) were obtained by site-directed mutagenesis of the wild-type *lspA* construct in pETDuet. Mutagenesis was performed using the Q5® Site-Directed Mutagenesis Kit (New England Biolabs) and specific primers are listed in Supplementary Table 5. The correct mutations were confirmed by sequencing (Eurofins Genomics). Expression and purification of all LspMrs wild-type and mutants for enzyme assays were performed as described above with the following modifications: TALON® Metal Affinity Resin (Clontech) was used for IMAC purification and the detergent DDM was used for membrane solubilisation and protein purification. All mutants were further purified by size-exclusion chromatography with the exception of N133A and N133Q, where expression levels were low and purification did not proceed beyond the IMAC step.

**Crystallisation.** Aliquots of purified LspMrs at 14–16 mg/mL in Buffer D were removed from storage at −70 °C and thawed on ice. A five-fold molar excess of the antibiotic globomycin or myxovirescin in dimethyl sulfoxide (DMSO) was added directly to the protein solution and incubated for 30 min on ice to allow binding. The lipid cubic phase (LCP) was made by mixing two volumes of the protein–antibiotic solution with three volumes of host lipid monoolein (NuChek) using two 100 μL gastight glass syringes (Hamilton) connected by a syringe narrow-bore coupler[31,32].

In meso crystallisation trials were set up by dispensing 50 nL boluses of the protein-laden mesophase onto a silanized 96-well glass crystallization plate (127.8 mm × 85.5 mm, 1 mm thickness, Marienfeld) to which a 96-well double stick spacer (77 mm × 112 mm, 140 μm thickness, Saunders Corporation) had been applied[33]. Each bolus was covered with 800 nL of precipitant solution using a Gryphon LCP (Art Robbins) or a Xantus crystallization robot (SIAS)[33]. Plates were sealed with a glass cover plate (77 mm × 112 mm, ~170 μm thickness, Marienfeld) and incubated in a Formulatrix Rock Imager (RockImager 1500, Formulatrix, USA) at 20 °C.

Best quality crystals of LspMrs with globomycin were obtained using precipitant solutions containing 47.1%(w/v) polyethylene glycol 1,000, 150 mM Tris/HCl pH 8.0 and 80 mM potassium bromide. The bipyramid-shaped crystals appeared after 4 days at 20 °C and continued to grow to a maximum size of $50 \times 50 \times 50$ μm$^3$ in 15 to 20 days (Supplementary Fig. 2c). Best quality crystals of LspMrs with myxovirescin were obtained using precipitant solutions containing 100 mM MES/NaOH pH 6.5, 40%(v/v) PEG400, 400 mM ammonium fluoride and 80 mM magnesium sulfate. The thin hexagon-shaped crystals appeared after 2–3 days and reached a maximum size of 80 μm in the longest dimension after 21 days (Supplementary Fig. 2d). Crystals from the lipid cubic phase were loop-harvested using MiTeGen cryoloops and snap-cooled in liquid nitrogen directly and without added cryo-protectant[32].

**Enzyme assays.** The peptidase activity of LspMrs and *P. aeruginosa* LspA (LspPae) was quantified in a coupled gel-shift assay developed previously for LspPae[5]. The *P. aeruginosa* lipoprotein pre-proICP and the first two enzymes in the bacterial lipoprotein processing pathway Lgt and LspA were expressed and purified, as described[5]. For FRET assays LspMrs was expressed and purified as described above and LspPae was expressed and purified as described in a published protocol[5].

Gel-shift activity assay. For time course assays, reactions were set up containing 12 μM pre-proICP, 250 μM DOPG (Avanti Polar Lipids, Inc.) and 1.2 μM Lgt in Buffer E (50 mM Tris/HCl pH 7.5, 150 mM NaCl, 1 mM DTT, 0.02%(w/v) LMNG). The assay mixtures were incubated at 37 °C and 200 rpm for 60 min to allow Lgt catalysed conversion of pre-proICP to the LspA substrate proICP. The LspA reaction was initiated by adding 0.5 μM LspA directly to the assay mixture. At timed intervals, 20 μL samples were removed and the reaction was stopped by adding 10 μL of 4x SDS loading buffer (62.5 mM Tris/HCl pH 6.8, 2.5%(w/v) SDS, 0.002%(w/v) bromophenol blue, 0.5 M β-mercaptoethanol, 10%(v/v) glycerol). To compare the activity of wild-type and mutant LspMrs, assays were allowed to proceed for 30 min before being stopped with SDS loading buffer. For dose response assays, globomycin or myxovirescin were added to the reaction mixture to a final concentration of 0 to 3.2 mM. The reactions were initiated by adding LspA and allowed to proceed for 30 min. After the reaction was stopped with SDS buffer, 10 μL aliquots were loaded on precast Mini-PROTEAN® TGX™ gels (Bio-Rad) and run with Tris/glycine buffer (25 mM Tris/HCl pH 8.0, 250 mM glycine, 0.1% (w/v) SDS). The gels were stained with InstantBlue™ (Expedeon) Coomassie based stain and were imaged using a Bio-Rad Gel-Doc imager (Supplementary Fig. 4). Band intensities of ICP were quantified using Image Lab. Dose-response plots to determine IC$_{50}$ values were generated using Prism® 5 software from GraphPad.

FRET activity assays. A FRET based assay was developed using a labelled lipopeptide substrate (Supplementary Methods, Supplementary Fig. 3b) to directly monitor the protease activity of LspA. The optimised LspMrs assay buffer included

100 mM Tris/HCl pH 7.8, 150 mM NaCl and 0.09%(w/v) DDM. The corresponding optimized buffer for LspPae included 100 mM MES/NaOH pH 5.6, 150 mM NaCl and 0.05%(w/v) LMNG. All assays were performed at 37 °C.

Reaction mixtures were prepared by pipetting the FRET lipopeptide substrate (dissolved in DMSO) directly into assay buffer at a final concentration of 80 μM. Reaction mixtures were prepared by vortexing (Clifton, cyclone CM-1) for 2 s and incubated at 37 °C, 800 rpm for 10 to 30 min. Equal volumes of the reaction mixture were pipetted into 1.5 mL Eppendorf tubes and 0 to 150 μM globomycin or myxovirescin (in DMSO) added. For dose-response assays, the final DMSO concentration in all reactions was 10%(v/v). Reaction mixtures were pipetted into wells of a 96- (Thermo Scientific) or a 384-well plate (4titude®) and incubated for 10 min at 37 °C in a SpectraMax M3 or a SpectraMax M5e plate reader (Molecular devices) prior to starting the assay. Assays were initiated by adding enzyme to a final concentration in the range 10–300 nM. Total reaction mixture volume was 50 μL. Reactions were monitored by fluorescence (Ex/Em, 320 nm/420 nm) for 30 to 60 min at 37 °C. Initial rates were calculated from the fluorescence intensity versus time progress curves (Supplementary Figs. 5 and 6). Occasionally, lag periods in the progress curves were observed (Supplementary Discussion). In such cases, initial rates were calculated after the lag period had passed (typically 1–5 min) and a steady rate had been established. Dose-response plots to determine IC$_{50}$ values were generated using Prism® 5 software from GraphPad.

It was possible to convert units of fluorescence change into moles of substrate consumed or products formed because pure products of LspA action on Abz-LALAGC*SS-nY-NH$_2$ had been synthesized and were available. These were used to create a standard curve of fluorescence versus moles of the two products in equimolar amounts and to calculate specific activity values. Inner filter effect correction was not required in the current study because absorbance values of samples used for FRET measurements never exceeded 0.06 and 0.02 at 320 and 420 nm, respectively, and the sum of absorbance at the excitation and emission wavelengths did not exceed 0.08[34]. A calibration curve generated using the purified FRET products showed that fluorescence was linear at the substrate concentrations used.

**LspMrs proICP docking.** Docking of LspMrs and prolipoprotein proICP was performed with HADDOCK (version 2.2 webserver)[35]. The model of prolipoprotein proICP docked into LspPae[5] and the LspMrs structure (6RYO) were used as docking models. In a text editor (Notepad, Microsoft), the atom record labels for the acyl chains were relabelled HETATM and the lipobox cysteine were renamed CYC. The files were uploaded to the HADDOCK webserver for docking. In HADDOCK, LspMrs residues 52-57 and 68-154, corresponding to H1, H2 and H3 and the EL were defined as active. Likewise, ProICP residues 1-22, corresponding to the signal peptide, were defined as active. All other residues were defined as passive. The HADDOCK distance restraints TBL file was uploaded. The target distance was set to 1.8 Å. The upper and lower distance margins were set to 0.1 Å. The top nine docking solutions (lowest HADDOCK scores) were downloaded and the docking solutions inspected manually in PyMOL. The top solution made the most sense by eye and was chosen as the model for inclusion in Supplementary Fig. 9.

**Globomycin production.** Bacterial strain and culture conditions. *Streptomyces hagronensis* strain 360, NRRL 15064[36] was used to produce globomycin. Spores were generated on ISP2 medium[37] agar plates at 30 °C for 10–14 days, and were preserved in 20% glycerol at −80 °C. For globomycin production a 1% spore inoculum was prepared in SHG-V (1% glycerol, 1% Bacto™ peptone, 1% Bacto™ yeast extract, distilled water, pH 7.0)[36] and was cultivated at 28 °C, 220 rpm for 3 days. It was further inoculated (5%) into SOYF6 medium (4.5% glycerol, 2% Bacto™ beef extract, 1% soy flour (Hensel), 0.2% CaCO$_3$, distilled water, pH 6.0 adjusted with KOH) (adapted from[20]) and incubated at 26 °C, 220 rpm for 5 days. In all, 2% amberlite XAD-16 was added after cultivation was complete.

Extraction and purification. The culture broth was centrifuged at $8300 \times g$ for 10 min at RT and DCM was added to the cell pellet. The DCM extract was concentrated and washed once with 500 mL 0.05 M HCl and 2% NaHCO$_3$ and twice with 200 mL of brine followed by drying with Na$_2$SO$_4$[20]. The organic phase was dried, fractionated on silica by flash chromatography using a linear gradient from 0 to 100% B using (A) hexane + 0.1% formic acid (FA) and (B) 80% EtOAc, 20% MeOH + 0.1% FA. Fractions containing globomycin were subjected to flash chromatography again this time using (A) EtOAc + 0.1% FA and (B) MeOH + 0.1% FA with a linear gradient from 1% to 20% B. Obtained samples were purified using preparative LC-MS (Dionex Ultimate 3000 coupled with Bruker High capacity ion trap mass spectrometer). Separation of the sample was carried out on a 5 μm EVO C18 100 Å LC column (250 mm × 10.0 mm, Kinetex) using a linear gradient (5–95% B in 31 min) from (A) ultrapure water + 0.1% FA to (B) acetonitrile + 0.1% FA and on a 5 μm Biphenyl 100 Å LC column (250 mm × 10.0 mm, Kinetex) a linear gradient of 42–43% B in 22 min at a flow rate of 5 mL/min and 45 °C.

Quantitation, HRMS and NMR structural analysis. An UHPLC system (UltiMate 3000 LC (Dionex) with 1.7 μm Acquity UPLC BEH C-18 Column (2.1 mm × 100 mm) coupled with an electrospray ionization (ESI) source linked to an amaZon speed MS (Bruker Daltonics; 3D ion trap)) was used for quantification. Separation of a 1 μL sample was achieved using a linear gradient (56–57% B in 7.5 min) from (A) ultrapure water + 0.1% FA to (B) acetonitrile + 0.1% FA at a

flow rate of 0.600 mL/min at 45 °C. For precise mass determination (HRMS) samples were measured on a UHPLC coupled to maXis 4 G MS system (Bruker Daltonics; Q-ToF). Separation of a 1 μL sample, with 1.7 μm Acquity UPLC BEH C-18 Column (2.1 mm × 50 mm), was achieved using a linear gradient (5–95% B in 6 min) from (A) ultrapure water + 0.1% FA to (B) acetonitrile + 0.1% FA at a flow rate of 0.600 mL/min at 45 °C. Absorption spectra (200-600 nm) were recorded by a diode array detector. Mass spectra were acquired in centroid mode ranging from 200 to 2000 m/z (amaZon) or from 150 to 2500 m/z (maXis 4 G) in positive ionization mode with auto MS-MS fragmentations. NMR spectra (1 H, 2D) were recorded on a Bruker Ascend 700 spectrometer with a 5 mm TXI cryoprobe (1 H at 700 MHz, 13 C at 175 MHz) at ambient temperature using standard pulse programs and results compared to the literature[38]. Final purity was estimated at >95% by LCMS, HRMS and NMR.

**Myxovirescin production**. Myxovirescin was produced by growing *Myxococcus virescens* strain Mx v48 in a 70 L fermenter, essentially as described by Gerth et al.[39] with the following modification: the medium was supplemented with 185 mL amberlite XAD-16 resin (Sigma Aldrich). Elution from the resin was performed using 800 mL methanol. After evaporating off the methanol, the dry material was extracted three times with ethyl acetate. The extract was dried using sodium sulfate followed by filtration and evaporation of the solvent. After resolubilisation in 200 mL methanol, a liquid-liquid extraction using 3 × 200 mL hexane was performed. The methanol was again evaporated yielding 750 mg of dry material. The extract was further separated using a column (70 mm × 600 mm) prepacked with Sephadex LH20 (Sigma Aldrich). The flow rate was 9 mL per min using methanol as the mobile phase. Fractions containing myxovirescin were pooled, dried, and the weight of the extract was determined to be 146 mg. Further purification was performed using RP-HPLC on a Nucleodur 100 EC C18 column (10 μm, 250 mm × 21 mm) and a flow rate of 20 mL per min with detection at 254 nm. The mobile phase consisted of solvent A (methanol:water, 1:1) and solvent B (methanol). A gradient of 30 min was applied with a steady increase of solvent B from 55% to 62%. The final yield of purified myxovirescin was 73 mg.

**Diffraction data collection and processing**. X-ray diffraction experiments were performed at the protein crystallography beamline X06SA-PXI, Swiss Light Source (SLS), Villigen-PSI, Switzerland. Data were collected at 100 K with a 10 × 10 μm² microfocused X-ray beam at 13.476 keV (0.92003 Å) for LspMrs-globomycin and at 12.398 keV (1 Å) for LspMrs-myxovirescin using SLS data acquisition software suites (DA+)[40]. Continuous grid-scans were used to locate crystals in cryogenically cooled mesophase samples[41]. Data were collected in steps of 0.2° at 0.1 s per step using the EIGER 16 M detector operated in continuous/shutterless data collection mode.

A 90° data set for LspMrs-globomycin, collected at a flux of $1.4 \times 10^{12}$ photons s$^{-1}$, was processed using *autoPROC/STARANISO*[42,43], scaled and merged with *XSCALE*[44]. The crystals, which belonged to space group $P3_221$, diffracted to 1.92 Å resolution with an ellipsoidal completeness of 62.5%, a $CC_{1/2}$ of 0.47 and an $I/\sigma$ of 1.8 in the highest resolution shell. The same flux was used for data collection with LspMrs-myxovirescin crystals. Two data sets of 60° and 90° for LspMrs-myxovirescin were processed with *XDS* and scaled and merged with *XSCALE*. The crystals, which belonged to space group $P6_122$, diffracted to 2.30 Å resolution with a completeness of 100%, a $CC_{1/2}$ of 0.22 and an $I/\sigma$ of 0.9 in the highest resolution shell. Data collection and processing statistics are provided in Table 2. Electron density defining the bound antibiotics is shown in Supplementary Fig. 11.

**Structure solution and refinement**. Molecular replacement (MR) was employed for phasing using a poly-Ala structure of 5DIR[5] as search model for LspMrs-globomycin and a solution was obtained using *Phaser*[45]. The MR solution was submitted together with the LspMrs sequence to *PHENIX-AutoBuild*[46] for initial model building. The 155 residues in LspMrs-globomycin were built and the structure was iteratively refined to $R_{work}/R_{free}$ factors of 0.25/0.28 using *Coot*[47] and *Phenix.refine*[48]. The structure of the LspMrs-globomycin was used as search model for LspMrs-myxovirescin and a solution was obtained using *Phaser*. The 164 residues in LspMrs were built and the structure was refined to $R_{work}/R_{free}$ factors of 0.25/0.28 using *Coot* and *Phenix.refine*, respectively. Refinement statistics are reported in Table 2. Figures of molecular structures were generated with *PyMOL*[49].

## Data availability

Data supporting the findings of this paper are available from the corresponding author upon reasonable request. A reporting summary for this Article is available as a Supplementary Information file. The structure and structure factors for LspMrs, LspA from *S. aureus* (MRSA), in complex with globomycin and myxovirescin A1 were deposited into the Protein Data Bank under accession codes 6RYO and 6RYP, respectively. The source data underlying Fig. 2 and Supplementary Figs. 1, 4, 5, 6a–d, 8, 18 and 19a are provided as a Source Data file.

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

## Acknowledgements

We thank D. Weichert for useful comments and discussions, for assistance with assay development and validation and the preparation of figures, P. Krawinski for performing docking studies, Lena Figur for assistance with protein production and purification, D. Muldowney for assistance with microbiological work, A. Horswill (University of Colorado) for the kind gift of the *S. aureus* LAC strain, E. Oueis for NMR analysis of globomycin, synchrotron facility scientists, M. Wang in particular, at the Swiss Light Source (X06SA and X10SA) and the Diamond Light Source (I24) for assistance and support, and past and present members of the Membrane Structural and Functional Biology group for assorted contributions to the study. Funding: This work was supported in part by Science Foundation Ireland award 16/IA/4435 (M.C.) and 15/CDA/3310 (E.M.S. and K.B.), an Irish Research Council fellowship GOIPG/2016/1238 (K.B.) and by the European Union's Horizon 2020 research and innovation program under the Marie-Skłodowska-Curie grant agreement No. 701647 (C-Y.H.).

## Author contributions

M.C. devised the overall research plan; J.B., S.O. and X.Y. produced and purified proteins and performed FRET and gel-shift assays; J.B. optimised the FRET assay; X.Y. and J.B performed in meso crystallisation screening; X.Y. crystallised the protein and optimised crystallisation conditions. S.O. performed site-directed mutagenesis. J.B., C-Y.H., S.O., X.Y. and V.O. collected X-ray data; C-Y.H. and V.O. solved structures; M.R. performed globomycin synthesis and purification; K.B. performed FRET lipopeptide synthesis and purification; J.G and M.Z. devised the approach to examine the contribution of LspA to MRSA infection, generated mutant and complemented mutant of MRSA and performed in vivo microbiological work; all authors analyzed data; S.O., R.M., E.M.S., J.A.G., V.O. and M.C. supervised research; M.C. wrote the paper with input from all authors.

## Competing interests

The authors declare no competing interests.
