## [Peer Review File · Nature Communications]

Reviewers' Comments:

Reviewer #1:

Remarks to the Author:

The manuscript by Olatunji et al describes the interaction of two macrocycle antibiotic natural products, globomycin and myxovirescin with their molecular target LspA. The latter is an Asp peptidase required for lipoprotein processing and is essential for life in gram negative bacteria and for virulence in some Gram positives. The 3D structures of the antibiotics in complex with LspA from MRSA were determined and despite the fact that the antibiotics share no structural similarities or common biosynthetic pathways, they are found to bridge the catalytic Asps through a common mechanism (via their primary hydroxyl groups) and occupy overlapping regions the active site through a shared 3D structure that the authors call the 'spine'. This finding is a remarkable example of convergent evolution towards a common 3D structure with biological impact. It also offers a structure that has the potential to guide downstream drug discovery studies.

Overall, I am very impressed by the scholarship in this study and its potential for impact. I would be very supportive of publication.

A few minor issues:

- 1) In Table 1, can these data be related to MIC? I realize that this may not be possible for the MRSA gene, but any measure of biological activity would be useful here I think to calibrate the IC50 values.
- 2) Do the HIV protease inhibitors in Fig 5 inhibit either LspA enzymes?
- 3) It would help to have a line (e.g. ChemDraw) addition to Fig 3 showing the Td structure of the complex.
- 4) The SS kinetic data in Fig S4 look to be more indicative of a substrate inhibition model than simple Michaelis-Menten kinetics. Have the authors explored this and compared fit statistics to both models?

Reviewer #2:

Remarks to the Author:

The manuscript by Olatunji et al., reports two high resolution crystal structures of the lipoprotein processing enzyme, LpsA, in complex with two antibiotic molecules. The impact of this work is clear, these structures provide high quality information on the interactions between two well-known antibiotic molecules. Additional interest is generated from the discovery that while these two antibiotic molecules are structurally dissimilar and have evolved from different bacterial species, they target the same binding site. This work thus lays solid ground for developing a new class of small molecule therapeutics against MRSA.

Overall the study is well written, and as usual from this group, thorough in the methods and analysis. I have no major concerns but a few comments and suggestions.

1. What evidence exists that LpsA processes substrates through a tetrahedral intermediate? Has this been shown through structural or kinetic analysis, such as isotope effects? It was unclear to me whether the authors interpretation comes that the antibiotics 'act as a non-cleavable tetrahedral intermediate was based on experimental evidence or conjecture?
2. It would helpful to put distances and residue IDs on all the Figures in the main paper.
3. Can the authors provide Polder maps for the ligands? These could go next to the map figures in the SI or in the main paper.

4. Can the authors provide more information in the section on convergent evolution. What is known about the evolutionary history of the two bacterial species *S. hagronensis* and *M. xanthus*. Are they separated by a long period of history, or did these species evolve and diverge at a similar time? It was unclear to me whether this could equally be divergent evolution from an ancestral molecule that targeted the LpsA active site and was picked up by these two species and evolved away.

Reviewer #3:

Remarks to the Author:

The manuscript has numerous major flaws that make it entirely unsuitable for publication in Nature Communications or any other first-tier journal. The authors present an interesting structural biology manuscript under the guise of a drug discovery project. While the globomycin and myxovirescin antibiotic share similar motifs, there is no content related to their antimicrobial activity. The authors did not include microbiological measurements of growth inhibition. Likewise, the authors claim built-in resistance hardness, but no data is given to substantiate this claim. Myxovirescin is also known as "TA" and the authors state that it possess desirable physiochemical properties. Yet, reference 18 notes that "the antibiotic fails to achieve significant blood levels even when directly injected into the blood". It is unclear to the reviewer what is desirable about this property. Finally, the title reads "lessons in drug discovery" whereas there are no lessons given. Only broad speculation on how to create a fantasy antibiotic. Aside from the fact that nature does not provide lessons, nature only provides data, this manuscript is replete with verbose language that is the epitome of hyperbole.

The claims may be novel, but the manuscript in its current form is only suitable for a second-tier structural biology journal. There is little interest in the community or wider fields. The evidence is convincing for a structural biology conclusion not for a broader interpretation of drug design. To reach this level, the new molecule needs to be synthesized and tested.

Other noteworthy shortcomings are...

1. The title "lessons in drug design" is vague. Authors can be more specific regarding the research of the paper on structural biochemistry of MRSA LspA interacting with the two antibiotics, which would also make it easier for readers and search engines toward the same interest.
2. There are many abbreviations throughout the whole paper, so an addition of a table of abbreviation is recommended.
3. In Fig. 1, LPE is missing in the figure description. I assume it's lyso phosphatidylethanolamine, is it lipo-?.
4. The introduction should provide more background information on LspA enzyme, globomycin, and myxovirescin since they are the three key components in this paper. For examples, why did LspA become the focus of the study. Is it the only target of globomycin/myxovirescin? (citations?) Background information of the two antibiotics?
5. Page 3, Results of "LspA is required for MRSA...", why is the "data not shown" for the loss of LspA activity affecting bacterial growth?
-In Fig. 2, clarify in the caption each bacterial strain labeled in the figure: LAC, LAC LspA, LAC LspA (pALC2073), and LAC LspA (pALC2073::lspA) . If it's a mutant, should a delta sign Δ lspA be used to clarify the missing gene?
-Why only 3 hr post-inoculation? Is 3 hr long enough for Staph aureus to fully grow? And does a reduction to 40% enough to conclude that "LspA is required for the survival of MRSA in blood" because resistant bacteria have different coping/defense mechanisms in different conditions and environments, and adaptation would usually require a longer time and/or small percentage of initial inoculum could thrive after they adapt to a new condition (in this case, lacking the gene lspA).
6. Page 4, the subsection of the Results "LspMrs-globomycin complex structure." is not consistently formatted as the previous subsection which is a full sentence/statement. Keep it

consistent!

7. Page 4, the second last sentence "LspMrs cluster around globomycin in the active and substrate binding sites in ways that make good biochemical sense." Is not clear. Define "good biochemical sense" or clarify it.

8. Fig. 3, define "PDB" in the caption?

9. Page 6, Table 1 is blurry and hard to spot the superscript labels in the table. What are the last 10 constructs in the table: N52A ... to D136N? the IC50 values shown are the average of duplicate data but how close are the two values? Standard errors? The value with an asterisk is only from a single measurement? Can it be repeated to have a second value for consistent data? Superscript "d" as "estimated by eye" is introducing human/personal bias into the data table. If it's below detection limit, it should be considered ND, not-determined.

10. The last sentence(s) of each section should provide a brief conclusion/summary of that section to strengthen the point the authors want to make. This is missing in the LspMrs-globomycin and LspMrs-myxovirescin complex structure subsections.

11. Page 6, "LspMrs-myxovirescin complex structure" paragraph starts with the intro material, which is informative, but should be placed in the introduction as mentioned in 4.

And since it's similar to the previous complex structure with globomycin, are there Km and Vmax?

12. Page 7, remove "striking" and "extraordinary"

13. Fig. 4, the image of the two superposed antibiotics look very alike. However, their structures are not exactly as a mirror image of each other as butterfly's wings, so the butterfly with outstretched wings in E. is a bit misleading.

14. Page 8, in Fig. 5, the last sentence in the caption "to the left and right" should be corrected. From left to right/ or from right to left?

15. Page 8, the second paragraph, again I do not agree with the analogy of a butterfly because it's misleading to the impression of chemical superimposed structures which is not in this case.

16. Page 9, EL flexibility: what is "EL" explaining the abbreviation in the text first and as mentioned making a table for all abbreviations will be helpful for readers.

17. For the whole Results section, the text can be more concise instead of repeating the description of the structure of each LspA-antibiotic complex. Each figure/table should also be discussed thorough and in depth in their order instead of going back and forth on each figure or table.

18. Discussion section, "Resistance hardy antibiotics" is confusing? The word "hardy" should be replaced with a better one.

19. The sentence, "At the same time however, mutating conserved residues will compromise the host organism as a result of ..." is not clear. What are "host organism" here? MRSA or human cells? And are there references to support this statement?

20. Page 10, convergent evolution is an interesting finding. Are there any findings on other antibiotics with convergent evolution?

21. Drug design is well written. However, LspA is essential in pseudomonas but not in MRSA. Yet, all the data shown is from interacting with MRSA LspA. And if it's not essential in MRSA, is it present in all staph aureus species or is it a disappearing gene. So then targeting it will make what benefits if LspA is not required?

22. Supplementary Materials: is all of the data regarding LspPae here from the reference (6)? Because most of its discussion cite that reference. Clarify more.

23. In Peptide synthesis, what are the 7 peptides for?

24. There is only one synthetic scheme for peptide 5? What about the other six?

25. Mass spectra and NMR spectra are shown only for peptide 1? Why? What about the other peptides?

20191007

A point-by-point response to the reviewers' comments follows.

Response to Reviewers' comments

Reviewer #1 (Remarks to the Author):

The manuscript by Olatunji et al describes the interaction of two macrocycle antibiotic natural products, globomycin and myxovirescin with their molecular target LspA. The latter is an Asp peptidase required for lipoprotein processing and is essential for life in gram negative bacteria and for virulence in some Gram positives. The 3D structures of the antibiotics in complex with LspA from MRSA were determined and despite the fact that the antibiotics share no structural similarities or common biosynthetic pathways, they are found to bridge the catalytic Asps through a common mechanism (via their primary hydroxyl groups) and occupy overlapping regions the active site through a shared 3D structure that the authors call the 'spine'. This finding is a remarkable example of convergent evolution towards a common 3D structure with biological impact. It also offers a structure that has the potential to guide downstream drug discovery studies.

Overall, I am very impressed by the scholarship in this study and its potential for impact. I would be very supportive of publication.

We thank the Reviewer for these favourable comments.

A few minor issues:

1) In Table 1, can these data be related to MIC? I realize that this may not be possible for the MRSA gene, but any measure of biological activity would be useful here I think to calibrate the IC50 values.

MIC and IC50 measurements are made in very different systems under very different conditions. Accordingly, the two don't necessarily correlate. Nonetheless, MIC values that have been reported in the literature have been included in footnotes to Table 1 for the benefit of the Reader.

2) Do the HIV protease inhibitors in Fig 5 inhibit either LspA enzymes?

We have not tested any of these inhibitors. It is part of a longer-term project.

3) It would help to have a line (e.g. ChemDraw) addition to Fig 3 showing the Td structure of the complex.

It is not clear what the Reviewer is requesting us to do here? Upon clarification, we will make every effort to address the point fully.

4) The SS kinetic data in Fig S4 look to be more indicative of a substrate inhibition model than simple Michaelis-Menten kinetics. Have the authors explored this and compared fit statistics to both models?

In fig. S5B, the highest concentration data point is below the best fit line for simple Michaelis-Menten kinetics. We have used a more complex model that includes substrate inhibition and the fit is a little better. This is expected; there are more adjustable parameters. A note to this effect has been included in the legend to fig. S5.

Solubility is an issue that is referred to in the manuscript under 'Complex nature of the LspA assay reaction mix'. This relates to solubility of the enzymes, the substrates, the products and the antibiotics used in the assay. While we would like to define substrate saturation more completely by going to considerably higher concentrations than are used in fig. S5, this is not possible with our current FRET lipopeptide. Other, more soluble substrates will be investigated as part of a separate project.

Reviewer #2 (Remarks to the Author):

The manuscript by Olatunji et al., reports two high resolution crystal structures of the lipoprotein processing enzyme, LpsA, in complex with two antibiotic molecules. The impact of this work is clear, these structures provide high quality information on the interactions between two well-known antibiotic molecules. Additional interest is generated from the discovery that while these two antibiotic molecules are structurally dissimilar and have evolved from different bacterial species, they target the same binding site. This work thus lays solid ground for developing a new class of small molecule therapeutics against MRSA.

Overall the study is well written, and as usual from this group, thorough in the methods and analysis. I have no major concerns but a few comments and suggestions.

We thank the Reviewer for these favourable comments.

1. What evidence exists that LpsA processes substrates through a tetrahedral intermediate? Has this been shown through structural or kinetic analysis, such as isotope effects? It was unclear to me whether the authors interpretation comes that the antibiotics 'act as a non-cleavable tetrahedral intermediate was based on experimental evidence or conjecture?

We have no direct experimental evidence that this is the mechanism used by LspA. It is conjecture based on what has been reported for other aspartyl proteases, such as HIV protease. Many of the several thousand inhibitors developed to target HIV protease include a pharmacophore that has been referred to as acting as a non-cleavable tetrahedral intermediate analog (ref. 15, Brik and Wong 2003). This same motif is found in globomycin and myxovirescin. It is for these reasons we refer to the enzyme as acting via a tetrahedral intermediate. We have added clarifications (lines 152, 381, 389) to this effect in the manuscript and thank the Reviewer for the comment.

2. It would helpful to put distances and residue IDs on all the Figures in the main paper.

Done.

3. Can the authors provide Polder maps for the ligands? These could go next to the map figures in the SI or in the main paper.

Polder-OMIT maps replace the original 2Fo-Fc maps in fig. S8. We thank the Reviewer for the suggestion.

4. Can the authors provide more information in the section on convergent evolution. What is known about the evolutionary history of the two bacterial species *S. hagnonensis* and *M. xanthus*. Are they separated by a long period of history, or did these species evolve and diverge at a similar time? It was unclear to me whether this could equally be divergent evolution from an ancestral molecule that targeted the LpsA active site and was picked up by these two species and evolved away.

We have modified the section on Convergent evolution (line 355) in response to this comment and to a comment (item 20) from Reviewer #3. This section is copied here:

Convergent evolution. Given that the organisms responsible for producing globomycin and myxovirescin come from different lineages of bacteria, the two antibiotics provide a convincing example of convergent evolution. Globomycin is produced by at least five *Streptomyces* species (23) belonging to the phylum, actinobacteria. *Streptomyces* are aerobic, Gram-positive, immotile filamentous bacteria containing branching mycelia and aerial hyphae where chains of spores form in a manner that resembles most filamentous fungi (24). By contrast, *Myxococcus xanthus*, the source organism for myxovirescin, belongs to the phylum, proteobacteria, and is a Gram-negative bacteria, known for its social behaviour, gliding motility, and fruiting body and myxospore formation (25). Both bacterial species are found in soil, a microbe-rich environment, where they use the products of secondary metabolism for defence. Otherwise, these two antibiotic producing microorganisms have very little in common. Indeed, they are as far apart in the kingdom bacteria as can be. All bacterial species contain the *LspA* gene for purposes of primary metabolism. However, there are only two bacterial genera, *Streptomyces* and *Myxococcus*, known to produce molecules that target *LspA*. There is no evidence of an ancestral molecule targeting the *LspA* active site.

Convergent evolution refers to the acquisition over time of similar form and function in organisms of distinct genetic origin. In the current example, that commonality finds expression in two entirely unrelated secondary metabolites. These have been crafted through the action of profoundly different multistep pathways in different bacterial species to target identically a specific catalytic dyad in the lipoprotein processing enzyme, *LspA*. Myxovirescin is synthesized from short chain carboxylates by a polyketide synthase (22). By contrast, a giant nonribosomal peptide synthetase is implicated in globomycin synthesis (Remškar and Müller, unpublished). Remarkably, the two antibiotics have achieved potency and specificity through a critical hydroxyl group that extends from profoundly different molecular scaffolds (Fig. 1). In myxovirescin, the scaffold is a macrolactam lactone. In globomycin, it is a cyclic depsipeptide. Therefore, the biosynthetic machineries in *Myxococcus* and *Streptomyces* have been able to fabricate molecular features in myxovirescin and globomycin, respectively, to mimic similar as well as distinct characteristics of the natural prolipoprotein substrate and the tetrahedral transition state intermediate it presumably forms during proteolytic cleavage.

Other examples of natural antibiotics that may have emerged through convergent evolution include rifampicin and sorangicin which target the β subunit pocket in RNA polymerase (RNAP) (26) and ripostatin and corallopyronin which bind to the switch region of RNAP (27).

Reviewer #3 (Remarks to the Author):

The manuscript has numerous major flaws that make it entirely unsuitable for publication in Nature Communications or any other first-tier journal. The authors present an interesting structural biology manuscript under the guise of a drug discovery project.

We very much appreciate the Reviewer's comments on our work and we are grateful for the thoughtful feedback provided. Like Reviewer #1 and Reviewer #2, we felt that the work, as presented, had significant merit in elucidating the mechanism of action of two natural antibiotics and in so doing at close to atomic resolution. The work clearly sets the stage for a drug design and discovery project. We have two 'hits' in hand and we know how they inhibit the enzyme in molecular detail. The spine of 19 atoms that overlap in the superposed complexes identifies a 'pharmacophore' that can be used to inform progression from hit to lead. This is part of an ongoing longer-term project.

While the globomycin and myxovirescin antibiotic share similar motifs, there is no content related to their antimicrobial activity. The authors did not include microbiological measurements of growth inhibition.

MIC values from the literature have now been included in the footnotes to Table 1. See also response to Reviewer #1, item 1.

Likewise, the authors claim built-in resistance hardiness, but no data is given to substantiate this claim.

Built-in resistance hardiness is a proposal based on an examination of the mechanism by which both antibiotics interact with LspA. We are not making a claim to this effect. To avoid any confusion, we referred to this as an hypothesis in the Discussion under 'Resistance hardy antibiotics' (line 334).

Myxovirescin is also known as "TA" and the authors state that it possess desirable physiochemical properties. Yet, references 18 notes that "the antibiotic fails to achieve significant blood levels even when directly injected into the blood". It is unclear to the reviewer what is desirable about this property.

Myxovirescin has been reported to have adhesive properties (refs. 20, 21). These make it attractive as an antibiotic for treating surfaces and interfaces, as are found in indwelling catheters and around teeth and bone. In the revised manuscript, 'physicochemical' has been replaced by 'adhesive' to avoid any confusion (line 208).

Finally, the title reads "lessons in drug discovery" whereas there are no lessons given. Only broad speculation on how to create a fantasy antibiotic. Aside from the fact that nature does not provide lessons, nature only provides data, this manuscript is replete with verbose language that is the epitome of hyperbole.

The word 'lessons' was used simply to convey the sense that these two profoundly different antibiotics which bind with similar 'spines' to a common target should inform a rational structural approach to the design and development of new therapeutics. There was no intent to exaggerate or engage in hyperbole. Rather it was to show how nature has evolved potent inhibitors to function in similar ways and how these structures should likewise provide a guide (a lesson) for the design of new inhibitors.

The claims may be novel, but the manuscript in its current form is only suitable for a second-tier structural biology journal. There is little interest in the community or wider fields.

With respect, we refer the Reviewer to the general comments provided by Reviewer #1 and Reviewer #2.

The evidence is convincing for a structural biology conclusion not for a broader interpretation of drug design. To reach this level, the new molecule needs to be synthesized and tested.

We agree with the Reviewer that new molecules need to be synthesized and tested. Such efforts are ongoing. However, they are beyond the scope of the current manuscript.

Other noteworthy shortcomings are...

1. The title "lessons in drug design" is vague. Authors can be more specific regarding the research of the paper on structural biochemistry of MRSA LspA interacting with the two antibiotics, which would also make it easier for readers and search engines toward the same interest.

As intimated by the Reviewer, we have modified the title as follows to make it more specific and informative:

Structure of lipoprotein signal peptidase II from *Staphylococcus aureus* in complex with antibiotics globomycin and myxovirescin

However, we would very much like to retain the original title and ask the Reviewer to consider the request in light of our responses above.

2. There are many abbreviations throughout the whole paper, so an addition of a table of abbreviation is recommended.

We appreciate this useful suggestion. A table of abbreviations is included in the Supplementary Text (Table S1).

3. In Fig. 1, LPE is missing in the figure description. I assume it's lyso phosphatidylethanolamine, is it lipo-?.

Indeed, LPE refers to lyso-phosphatidylethanolamine. This has been entered in the legend to Fig. 1.

4. The introduction should provide more background information on LspA enzyme, globomycin, and myxovirescin since they are the three key components in this paper. For examples, why did LspA become the focus of the study. Is it the only target of globomycin/myxovirescin? (citations?) Background information of the two antibiotics?

Background information on LspA and the two antibiotics is provided in the introductory paragraphs along with the reasons for choosing LspA as a target. Providing additional detail in the introduction would mean that information more relevant in later sections would need to be repeated. With respect, we prefer to keep the introduction in its original form.

5. Page 3, Results of “LspA is required for MRSA...”, why is the “data not shown” for the loss of LspA activity affecting bacterial growth?

We now include data showing bacterial growth in plasma from the same blood donors (Fig. 2B) demonstrating that the defect in survival of MRSA in blood is due to killing by the cellular component and not due to a growth defect. Similarly, we now include data showing that wild-type MRSA252 and the *lspA* mutant show a similar profile of growth in rich media (TSB) (fig. S1). This is consistent with our assertion that inhibition of LspA is a strategy to reduce survival and proliferation of the bacterium *in vivo*.

-In Fig. 2, clarify in the caption each bacterial strain labeled in the figure: LAC, LAC *lspA*, LAC *lspA* (pALC2073), and LAC *lspA* (pALC2073::*lspA*) . If it's a mutant, should a delta sign Δ *lspA* be used to clarify the missing gene?

The mutant used in the bacterial growth study carries an inactivating transposon insertion in the *lspA* gene and not a deletion of the *lspA* gene. We used a standard complementation strategy introducing the gene back into the *lspA*-defective mutant background with a multicopy plasmid where the phenotype was reversed confirming that the phenotype is gene-dependent.

-Why only 3 hr post-inoculation? Is 3 hr long enough for Staph aureus to fully grow? And does a reduction to 40% enough to conclude that “LspA is required for the survival of MRSA in blood” because resistant bacteria have different coping/defense mechanisms in different conditions and environments, and adaptation would usually require a longer time and/or small percentage of initial inoculum could thrive after they adapt to a new condition (in this case, lacking the gene *lspA*).

The inclusion of data relating to bacterial growth in plasma (Fig. 2B) shows that, under these conditions, a 3-hour incubation period allows sufficient time for this MRSA strain to grow. As the experiment was performed *ex vivo* with freshly drawn hirudin-treated blood to preserve the activity of immune responses, including complement and phagocytes, time was important. The procedure we followed had been optimized previously and a 3-hour incubation period allows differences in bacterial fitness under the immune pressure to be detected. We consider the fitness reduction of 40% to be quite astonishing given the vast array of secreted and cell-bound immune evasion factors produced by this pathogen. However, we accept that the statement “LspA is required for the survival of MRSA in blood” could be misleading. Instead, we revise the relevant section (line 102) to read “These results show that LspA activity is important for the survival of MRSA in human blood. As a control, the same strains and mutants were incubated in fresh plasma from the same blood donors. There was no reduction in the ability of the *lspA* mutant to grow in human plasma (Fig. 2B) under the same conditions. This indicates that the *lspA* mutant has a reduced ability to survive killing by phagocytes in human blood rather than having a growth defect under the conditions used. These findings identified the LspA enzyme as an attractive target for treatment of MRSA infection.”

6. Page 4, the subsection of the Results “LspMrs-globomycin complex structure.” is not

consistently formatted as the previous subsection which is a full sentence/statement. Keep it consistent!

The heading has been modified for consistency following the suggestion of the Reviewer.

7. Page 4, the second last sentence “LspMrs cluster around globomycin in the active and substrate binding sites in ways that make good biochemical sense.” Is not clear. Define “good biochemical sense” or clarify it.

By this we mean that the conserved residues are all positioned close to (clustered around) the bound antibiotic which, in turn, has its blocking hydroxy between the catalytic dyad side chain carboxyls. Presumably, these residues play a role in enabling LspA to function biochemically as an enzyme that catalyses a critical step in lipoprotein posttranslational modification.

8. Fig. 3, define “PDB” in the caption?

Done.

9. Page 6, Table 1 is blurry and hard to spot the superscript labels in the table. What are the last 10 constructs in the table: N52A ... to D136N?

A revised version of the Table 1 has been included.

the IC₅₀ values shown are the average of duplicate data but how close are the two values? Standard errors? The value with an asterisk is only from a single measurement? Can it be repeated to have a second value for consistent data?

Standard errors have been included in Table 1. The entry with an asterisk has been replaced by an average and standard errors based on a repeat measurement, as suggested by the Reviewer.

Superscript “d” as “estimated by eye” is introducing human/personal bias into the data table. If it's below detection limit, it should be considered ND, not-determined.

“Estimated by eye.” have been removed. A note to the effect that the signal was too weak to be quantified remains in Table 1, footnote f.

10. The last sentence(s) of each section should provide a brief conclusion/summary of that section to strengthen the point the authors want to make. This is missing in the LspMrs-globomycin and LspMrs-myxovirescin complex structure subsections.

The recommendation made by the Reviewer has been followed for the LspMrs-globomycin complex structure subsection. This is not necessary for the LspMrs-myxovirescin subsection because the subsequent subsection immediately and logically extends the storyline introduced in the LspMrs-myxovirescin subsection.

11. Page 6, “LspMrs-myxovirescin complex structure” paragraph starts with the intro material, which is informative, but should be placed in the introduction as mentioned in 4.

The content of the Introduction has been addressed in item 4 above.

And since it's similar to the previous complex structure with globomycin, are there K_m and V_{max} ?

It is not clear what the Reviewer is requesting us to do here? Upon clarification, we will make every effort to address the point fully.

12. Page 7, remove “striking” and “extraordinary”

Done.

13. Fig. 4, the image of the two superposed antibiotics look very alike. However, their structures are not exactly as a mirror image of each other as butterfly's wings, so the butterfly with outstretched wings in E. is a bit misleading.

The analogy was made simply to convey a sense that the two antibiotics in the superpose splay apart like wings of a butterfly. We feel that referring to the likeness as ‘gives the impression’ (lines 259, 286) in no way implies mirror symmetry and should avoid confusion.

14. Page 8, in Fig. 5, the last sentence in the caption “to the left and right” should be corrected. From left to right/ or from right to left?

The statement is accurate and correct as written. It has not been changed.

15. Page 8, the second paragraph, again I do not agree with the analogy of a butterfly because it's misleading to the impression of chemical superimposed structures which is not in this case.

See response to item 13 above.

16. Page 9, EL flexibility: what is “EL” explaining the abbreviation in the text first and as mentioned making a table for all abbreviations will be helpful for readers.

This refers to the extracellular loop. It was introduced in line 147. It is now included in the table of abbreviations, Table S1.

17. For the whole Results section, the text can be more concise instead of repeating the description of the structure of each LspA-antibiotic complex. Each figure/table should also be discussed thorough and in depth in their order instead of going back and forth on each figure or table.

We have modified the text slightly to make it flow logically and to eliminate unnecessary repetition.

18. Discussion section, “Resistance hardy antibiotics” is confusing? The word “hardy” should be replaced with a better one.

There is reference in the literature to ‘Resistance-Resistant Antibiotics’ (Trends Pharmacol Sci. 35: 664–674, 2014). We had considered using this title but, on reflection, chose the adjective ‘hardy’ instead. It is used here in the same way that

'drought tolerant plants' are referred to as being 'drought hardy'. 'Resistant' is too strong and absolute an adjective. Respectfully, we prefer to retain our reference to the quality descriptor 'resistance-hardy'.

19. The sentence, "At the same time however, mutating conserved residues will compromise the host organism as a result of ..." is not clear. What are "host organism" here? MRSA or human cells? And are there references to support this statement?

By host, we refer to the bacterial cell. This is clarified in the revised version of the manuscript (line 338). Table 1 shows that mutating conserved residues affects the activity of LspA. Fig. 2 shows that mutating the *lspA* gene compromises *S. aureus* survival in whole human blood.

20. Page 10, convergent evolution is an interesting finding. Are there any findings on other antibiotics with convergent evolution?

Addressed fully in response to Reviewer #2 (item 4).

21. Drug design is well written. However, LspA is essential in pseudomonas but not in MRSA. Yet, all the data shown is from interacting with MRSA LspA. And if it's not essential in MRSA, is it present in all staph aureus species or is it a disappearing gene. So then targeting it will make what benefits if LspA is not required?

The *lspA* gene is present in all strains of *S. aureus*. It is not required for growth in laboratory media (fig. S1) or in blood plasma (Fig. 2B). However, our data convincingly shows that interfering with the activity of LspA (in this case mutation of the *lspA* gene) results in a significant defect in the ability of MRSA to survive in whole human blood. Thus, while the gene is not *essential* our findings suggest that it contributes to the pathogenesis of *S. aureus* infection by promoting the survival of the bacteria in whole human blood. Therefore, a major conclusion of this work is that targeting LspA is a host environment-specific strategy. Drugs that target LspA will therefore not be antibiotics but instead anti-virulence agents that diminish the ability of the pathogen to cause infection. An important benefit is that since anti-virulence agents do not kill the bacteria, resistant mutants are less likely to emerge. Therefore, we have identified LspA as a promising new target for MRSA treatments.

22. Supplementary Materials: is all of the data regarding LspPae here from the reference (6)? Because most of its discussion cite that reference. Clarify more.

All the data in the Supplementary Materials is new. It refers back to our earlier work on LspA from *P. aeruginosa* reported in ref. 6.

23. In Peptide synthesis, what are the 7 peptides for?

Under peptide synthesis, the other structures shown are intermediates prepared during the synthesis of the FRET probe. This is now clarified (line 967).

24. There is only one synthetic scheme for peptide 5? What about the other six?

See reply to item 23 above

25. Mass spectra and NMR spectra are shown only for peptide 1? Why? What about the other peptides?

Since the other structures are known compounds, we have not included full characterization data for them. This is standard practice in publications concerning organic synthesis.

Reviewers' Comments:

Reviewer #1:

Remarks to the Author:

The authors have addressed my questions adequately

Reviewer #2:

Remarks to the Author:

The authors have addressed the main queries I had with the original version. I think the current work presents novel insights into how two important antibiotic classes target LspA.

Reviewer #3:

Remarks to the Author:

The revised manuscript, as expected, is greatly improved. The authors addressed technical deficiencies and provided new information. However, some of this new information was obtained from literature reports by other laboratories rather than their own experimental studies. The manuscript is a fine example of structural biology. However, the central critique that work is premature for publication in Nature Com remains. The manuscript is unlikely to inspire vigor in other labs to create an organic butterfly with two different wings. This approach is not innovative and is an incremental advance in the field. If, however, the authors were able to synthesize this fantasy molecule (which is planned) and the fantasy molecule shows activity (testing is also planned), then the work would rise to the level of publishing in Nature.

20191102

A point-by-point response to the reviewers' comments follows.

Response to Reviewers' comments

Reviewer #1 (Remarks to the Author):

The authors have addressed my questions adequately

We thank the Reviewer for the time and effort devoted to providing a very useful critique of our work.

Reviewer #2 (Remarks to the Author):

The authors have addressed the main queries I had with the original version. I think the current work presents novel insights into how two important antibiotic classes target LspA.

We thank the Reviewer for the time and effort devoted to providing a very useful critique of our work.

Reviewer #3 (Remarks to the Author):

The revised manuscript, as expected, is greatly improved. The authors addressed technical deficiencies and provided new information. However, some of this new information was obtained from literature reports by other laboratories rather than their own experimental studies. The manuscript is fine example of structural biology. However, the central critique that work is premature for publication in Nature Com remains. The manuscript is unlikely to inspire vigor in other labs to create an organic butterfly with two different wings. This approach is not innovative and is an incremental advance in the field. If, however, the authors were able to synthesize this fantasy molecule (which is planned) and the fantasy molecule shows activity (testing is also planned), then the work would rise to the level of publishing in Nature.

We thank the Reviewer for the time and effort devoted to providing a very useful critique of our work. As noted in our original rebuttal, the butterfly analogy is just that, a likeness. A bicyclic compound inspired by the superposition of globomycin and myxovirescin in the two solved complex structures is again just a proposal that, with the proper resources, might be worth exploring. Relatedly, it has not escaped our notice that the flatness of the two cyclic antibiotics where they contact the LspA enzyme is reminiscent of ligands being developed against so-called undruggable targets with relatively featureless binding surfaces rather than pockets (Itoh & Inoue, Chem. Rev. 2019, 119, 10002–10031).